# Combinatorial Bandits with Relative Feedback

**Aadirupa Saha**
Indian Institute of Science, Bangalore
aadirupa@iisc.ac.in

**Aditya Gopalan**
Indian Institute of Science, Bangalore
aditya@iisc.ac.in

## Abstract

We consider combinatorial online learning with subset choices when only relative feedback information from subsets is available, instead of bandit or semi-bandit feedback which is absolute. Specifically, we study two regret minimisation problems over subsets of a finite ground set $[n]$, with subset-wise relative preference information feedback according to the Multinomial logit choice model. In the first setting, the learner can play subsets of size bounded by a maximum size and receives top-$m$ rank-ordered feedback, while in the second setting the learner can play subsets of a fixed size $k$ with a full subset ranking observed as feedback. For both settings, we devise instance-dependent and order-optimal regret algorithms with regret $O(\frac{n}{m} \ln T)$ and $O(\frac{n}{k} \ln T)$, respectively. We derive fundamental limits on the regret performance of online learning with subset-wise preferences, proving the tightness of our regret guarantees. Our results also show the value of eliciting more general top-$m$ rank-ordered feedback over single winner feedback ($m = 1$). Our theoretical results are corroborated with empirical evaluations.

## 1 Introduction

Online learning over subsets with absolute or *cardinal* utility feedback is well-understood in terms of statistically efficient algorithms for bandits or semi-bandits with large, combinatorial subset action spaces [15, 31]. In such settings the learner aims to find the subset with highest value, and upon testing a subset observes either noisy rewards from its constituents or an aggregate reward. In many natural settings, however, information obtained about the utilities of alternatives chosen is inherently relative or *ordinal*, e.g., recommender systems [25, 34], crowdsourcing [16], multi-player game ranking [22], market research and social surveys [9, 6, 24], and in other systems where humans are often more inclined to express comparative preferences.

The framework of dueling bandits [43, 46] represents a promising attempt to model online optimisation with *pairwise* preference feedback. However, our understanding of the more general and realistic online learning setting of combinatorial subset choices and *subset-wise* feedback is relatively less developed than the case of observing absolute, subset-independent reward information.

In this work, we consider a generalisation of the dueling bandit problem where the learner, instead of choosing only two arms, selects a subset of (up to) $k \geq 2$ many arms in each round. The learner subsequently observes as feedback a rank-ordered list of $m \geq 1$ items from the subset, generated probabilistically according to an underlying subset-wise preference model – in this work the Plackett-Luce distribution on rankings based on the multinomial logit (MNL) choice model [7] – in which each arm has an unknown positive value. Simultaneously, the learner earns as reward the average value of the subset played in the round. The goal of the learner is to play subsets to minimise its cumulative regret with respect to the subset with highest value.

Achieving low regret with subset-wise preference feedback is relevant in settings where deviating from choosing an optimal subset of alternatives comes with a cost (driven by considerations like revenue) even during the learning phase, but where the feedback information provides purely relative feedback. For instance, consider a beverage company that experimentally develops several variants of a drink (arms or alternatives), a best-selling subset of which it wants to learn to put up in the open market by trial and error. Each time a subset of items is put up, in parallel the company elicits relative preference feedback about the subset from, say, a team of expert tasters or through crowdsourcing.

The value of a subset can be modelled as the average value of items in it, which is however not directly observable, it being function of the open market response to the offered subset. The challenge thus lies in optimizing the subset selection over time by observing only relative preferences (made precise by the notion of Top-$k$-regret, Section 2.2).

A challenging feature of this problem, with subset plays and relative feedback, is the combinatorially large action and feedback space, much like those in combinatorial bandits [14, 19]. The key question here is whether (and if so, how) structure in the subset choice model – defined compactly by only a few parameters (as many as the number of arms) – can be exploited to give algorithms whose regret does not explode combinatorially. The contributions of this paper are:

**(1).** We consider the problem of regret minimisation when subsets of items $\{1, \ldots, n\}$ of size at most $k$ can be played, top $m \leq k$ rank-ordered feedback is received according to the MNL model, and the value of a subset is the mean MNL-parameter value of the items in the subset. We propose an upper confidence bound (UCB)-based algorithm, with a new max-min subset-building rule and a lightweight space requirement of tracking $O(n^2)$ pairwise item estimates, showing that it enjoys instance-dependent regret in $T$ rounds of $O(\frac{n}{m} \ln T)$. This is shown to be order-optimal by exhibiting a lower bound of $\Omega(\frac{n}{m} \ln T)$ on the regret for any *No-regret* algorithm. Our results imply that the optimal regret does not vary with the maximum subset size ($k$) that can be played, but improves multiplicatively with the length of top $m$-rank-ordered feedback received per round (Sec. 3).

**(2).** We consider a related regret minimisation setting in which subsets of size exactly $k$ must be played, after which a ranking of the $k$ items is received as feedback, and where the zero-regret subset consists of the $k$ items with the highest MNL-parameter values. In this case, our analysis reveals a fundamental lower bound on regret of $\Omega(\frac{n-k}{k\Delta_{(k)}} \ln T)$, where the problem complexity now depends on the parameter difference between the $k^{th}$ and $(k+1)^{th}$ best item of the MNL model. We follow this up with a subset-playing algorithm (Alg. 3) for this problem – a recursive variant of the earlier UCB-based algorithm – with a matching, optimal regret guarantee of $O\left(\frac{(n-k)\ln T}{k\Delta_{(k)}}\right)$ (Sec. 4).

We also provide extensive numerical evaluations supporting our theoretical findings. Due to space constraints, a discussion on related work appears in the Appendix.

## 2 Preliminaries and Problem Statement

**Notation.** We denote by $[n]$ the set $\{1, 2, ..., n\}$. For any subset $S \subseteq [n]$, we let $|S|$ denote the cardinality of $S$. When there is no confusion about the context, we often represent (an unordered) subset $S$ as a vector (or ordered subset) $S$ of size $|S|$ according to, say, a fixed global ordering of all the items $[n]$. In this case, $S(i)$ denotes the item (member) at the $i$th position in subset $S$. For any ordered set $S$, $S(i : j)$ denotes the set of items from position $i$ to $j$, $i < j, \forall i, j \in [|S|]$. $\mathbf{\Sigma}_S = \{\sigma \mid \sigma$ is a permutation over items of $S\}$, where for any permutation $\sigma \in \Sigma_S$, $\sigma(i)$ denotes the element at the $i$-th position in $\sigma, i \in [|S|]$. We also denote by $\mathbf{\Sigma}_S^m$ the set of permutations of any $m$-subset of $S$, for any $m \in [k]$, i.e. $\Sigma_S^m := \{\Sigma_{S'} \mid S' \subseteq S, |S'| = m\}$. $\mathbf{1}(\varphi)$ is generically used to denote an indicator variable that takes the value 1 if the predicate $\varphi$ is true, and 0 otherwise. $Pr(A)$ is used to denote the probability of event $A$, in a probability space that is clear from the context.

**Definition 1** (Multinomial logit probability model)**.** A Multinomial logit (MNL) probability model $\text{MNL}(n, \boldsymbol{\theta})$, specified by positive parameters $(\theta_1, \ldots, \theta_n)$, is a collection of probability distributions $\{Pr(\cdot|S) : S \subset [n], S \neq \emptyset\}$, where for each non-empty subset $S \subseteq [n]$, $Pr(i|S) = \frac{\theta_i \mathbf{1}(i \in S)}{\sum_{j \in S} \theta_j}$ $\forall 1 \leq i \leq n$. The indices $1, \ldots, n$ are referred to as 'items' or 'arms' .

**(i).** *Best-Item***:** Given an $\text{MNL}(n, \boldsymbol{\theta})$ instance, we define the *Best-Item* $a^* \in [n]$, to be the item with highest MNL parameter if such a unique item exists, i.e. $a^* := \arg\max_{i \in [n]} \theta_i$.

**(ii).** *Top-$k$ Best-Items***:** Given any instance of $\text{MNL}(n, \boldsymbol{\theta})$ we define the *Top-$k$ Best-Items* $S_{(k)} \subseteq [n]$, to be the set of $k$ distinct items with highest MNL parameters if such a unique set exists, i.e. for any pair of items $i \in S_{(k)}$ and $j \in [n] \setminus S_{(k)}$, $\theta_i > \theta_j$, such that $|S_{(k)}| = k$. For this problem, we assume $\theta_1 \geq \theta_2 \geq \ldots \theta_k > \theta_{k+1} \geq \ldots \geq \theta_n$, implying $S_{(k)} = [k]$. We also denote $\Delta_{(k)} = \theta_k - \theta_{k+1}$.

### 2.1 Feedback models

An *online learning algorithm* interacts with a $\text{MNL}(n, \boldsymbol{\theta})$ probability model over $n$ items as follows. At each round $t = 1, 2, \ldots$, the algorithm plays a subset $S_t \subseteq [n]$ of (distinct) items, with $|S_t| \leq k$, upon which it receives stochastic feedback defined as:

1. **Winner Feedback:** In this case, the environment returns a single item $J$ drawn independently from probability distribution $Pr(\cdot|S)$, i.e., $Pr(J = j|S) = \frac{\theta_j}{\sum_{\ell \in S} \theta_\ell} \, \forall j \in S$.

2. **Top-$m$-ranking Feedback** ($1 \leq m \leq k-1$)**:** Here, the environment returns an ordered list of $m$ items sampled without replacement from the MNL$(n, \boldsymbol{\theta})$ probability model on $S$. More formally, the environment returns a partial ranking $\boldsymbol{\sigma} \in \boldsymbol{\Sigma}_S^m$, drawn from the probability distribution $Pr(\boldsymbol{\sigma} = \sigma|S) = \prod_{i=1}^m \frac{\theta_{\sigma^{-1}(i)}}{\sum_{j \in S \setminus \sigma^{-1}(1:i-1)} \theta_j}, \; \sigma \in \boldsymbol{\Sigma}_S^m$. This can also be seen as picking an item $\boldsymbol{\sigma}^{-1}(1) \in S$ according to *Winner Feedback* from $S$, then picking $\boldsymbol{\sigma}^{-1}(2)$ from $S \setminus \{\boldsymbol{\sigma}^{-1}(1)\}$, and so on, until all elements from $S$ are exhausted. When $m = 1$, Top-$m$-ranking Feedback is the same as Winner Feedback. To incorporate sets with $|S| < m$, we set $m = \min(|S|, m)$. Clearly this model reduces to Winner Feedback for $m = 1$, and a full rank ordering of the set $S$ when $m = |S| - 1$.

## 2.2 Decisions (Subsets) and Regret

We consider two regret minimisation problems in terms of their decision spaces and notions of regret:

**(1). Winner-regret:** This is motivated by learning to identify the *Best-Item* $a^*$. At any round $t$, the learner can play sets of size $1, \ldots, k$, but is penalised for playing any item other than $a^*$. Formally, we define the learner's instantaneous regret at round $t$ as $r_t^1 = \sum_{i \in S_t} \frac{(\theta_{a^*} - \theta_i)}{|S_t|}$, and its cumulative regret from $T$ rounds as $R_T^1 = \sum_{t=1}^T r_t^1 = \sum_{t=1}^T \left( \sum_{i \in S_t} \frac{(\theta_{a^*} - \theta_i)}{|S_t|} \right)$,

The learner aims to play sets $S_t$ to keep the regret as *low* as possible, i.e., to play only the singleton set $S_t = \{a^*\}$ over time, as that is the only set with $0$ regret. The instantaneous Winner-regret can be interpreted as a shortfall in value of the played set $S_t$ with respect to $\{a^*\}$, where the value of a set $S$ is simply the mean parameter value $\frac{\sum_{i \in S} \theta_i}{|S|}$ of its items.

**Remark 1.** Assuming $\theta_{a^*} = 1$ (we can do this without loss of generality since the MNL model is positive scale invariant, see Defn. 1), it is easy to note that for any item $i \in [n] \setminus \{a^*\}$ $p_{a^*i} := Pr(a^*|\{a^*, i\}) = \frac{\theta_{a^*}}{\theta_{a^*} + \theta_i} \geq \frac{1}{2} + \frac{\theta_{a^*} - \theta_i}{4}$ (as $\theta_i < \theta_{a^*}, \forall i$). Consequently, the Winner-regret as defined above, can be further bounded above (up to constant factors) as $\tilde{R}_T^1 = \sum_{t=1}^T \sum_{i \in S_t} \frac{(p_{a^*i} - \frac{1}{2})}{|S_t|}$, which, for $k = 2$, is standard dueling bandit regret [45, 47, 42].

**Remark 2.** An alternative notion of instantaneous regret is the shortfall in the preference probability of the best item $a^*$ in the selected set $S_t$, i.e., $\tilde{r}_t^1 = \sum_{i \in S_t} \left( Pr(a^*|S_t \cup \{a^*\}) - Pr(i|S_t \cup \{a^*\}) \right)$. However, if all the MNL parameters are bounded, i.e., $\theta_i \in [a, b], \forall i \in [n]$, then $\frac{1}{b} \left( \sum_{i \in S_t} \frac{(\theta_{a^*} - \theta_i)}{|S_t|+1} \right) \leq \tilde{r}_t^1 \leq \frac{1}{a} \left( \sum_{i \in S_t} \frac{(\theta_{a^*} - \theta_i)}{|S_t|+1} \right)$, implying that these two notions of regret, $r_t^1$ and $\tilde{r}_t^1$, are only constant factors apart.

**(2). Top-$k$-regret:** This setting is motivated by learning to identify the set of *Top-$k$ Best-Items* $S_{(k)}$ of the MNL$(n, \boldsymbol{\theta})$ model. Correspondingly, we assume that the learner can play sets of $k$ distinct items at each round $t \in [T]$. The instantaneous regret of the learner, in this case, in the $t$-th round is defined to be $r_t^k = \left( \frac{\theta_{S_{(k)}} - \sum_{i \in S_t} \theta_i}{k} \right)$, where $\theta_{S_{(k)}} = \sum_{i \in S_{(k)}} \theta_i$. Consequently, the cumulative regret of the learner at the end of round $T$ becomes $R_T^k = \sum_{t=1}^T r_t^k = \sum_{t=1}^T \left( \frac{\theta_{S_{(k)}} - \sum_{i \in S_t} \theta_i}{k} \right)$. As with the Winner-regret, the Top-$k$-regret also admits a natural interpretation as the shortfall in *value* of the set $S_t$ with respect to the set $S_{(k)}$, with value of a set being the mean $\theta$ parameter of its arms.

## 3 Minimising Winner-regret

We first consider the problem of minimising Winner-regret. We start by analysing a regret lower bound for the problem, followed by designing an optimal algorithm with matching upper bound.

### 3.1 Fundamental lower bound on Winner-regret

Along the lines of [32], we define the following consistency property of any reasonable online learning algorithm in order to state a fundamental lower bound on regret performance.

**Definition 2** (*No-regret* algorithm). An online learning algorithm $\mathcal{A}$ is defined to be a *No-regret* algorithm for Winner-regret if for each problem instance $\text{MNL}(n, \boldsymbol{\theta})$, the expected number of times $\mathcal{A}$ plays any suboptimal set $S \subseteq [n]$ is sublinear in $T$, i.e., $\forall S \neq \arg\max_i \theta_i : \mathbf{E}_{\boldsymbol{\theta}}[N_S(T)] = o(T^{\alpha})$, for some $\alpha \in [0, 1]$ (potentially depending on $\boldsymbol{\theta}$), where $N_S(T) := \sum_{t=1}^{T} \mathbf{1}(S_t = S)$ is the number plays of set $S$ in $T$ rounds. $\mathbf{E}_{\boldsymbol{\theta}}[\cdot]$ denotes expectation under the algorithm and $\text{MNL}(n, \boldsymbol{\theta})$ model.

**Theorem 3** (Winner-regret Lower Bound). *For any No-regret learning algorithm $\mathcal{A}$ for Winner-regret that uses Winner Feedback, and for any problem instance MNL($n, \boldsymbol{\theta}$) s.t. $a^* = \underset{i \in [n]}{\arg\max} \theta_i$,*

*the expected regret incurred by $\mathcal{A}$ satisfies* $\underset{T \to \infty}{\lim\inf} \mathbf{E}_{\boldsymbol{\theta}}\left[ \frac{R_T^1(\mathcal{A})}{\ln T} \right] \geq \dfrac{\theta_{a^*}}{\left( \underset{i \in [n] \setminus \{a^*\}}{\min} \frac{\theta_{a^*}}{\theta_i} - 1 \right)} (n-1).$

**Note:** This is a problem-dependent lower bound with $\theta_{a^*} \left( \underset{i \in [n] \setminus \{a^*\}}{\min} \frac{\theta_{a^*}}{\theta_i} - 1 \right)^{-1}$ denoting a complexity or hardness term ('gap') for regret performance under any *'reasonable learning algorithm'*.

**Remark 3.** The result suggests the regret rate with only Winner Feedback cannot improve with $k$, uniformly across all problem instances. Rather strikingly, there is no reduction in hardness (measured in terms of regret rate) in learning the *Best-Item* using Winner Feedback from large ($k$-size) subsets as compared to using pairwise (dueling) feedback ($k = 2$). It could be tempting to expect an improved learning rate with subset-wise feedback as the number of items being tested per iteration is more ($k \geq 2$), so information-theoretically one may expect to 'learn more' about the underlying model per subset query. On the contrary, it turns out that it is intuitively 'harder' for a good (i.e., near-optimal) item to prove its competitiveness in just a single winner draw against a large population of its $k - 1$ other competitors, as compared to winning over just a single competitor for $k = 2$ case.

**Proof sketch.** The proof of the result is based on the change of measure technique for bandit regret lower bounds presented by, say, Garivier et al. [21], that uses the information divergence between two nearby instances $\text{MNL}(n, \boldsymbol{\theta})$ (the original instance) and $\text{MNL}(n, \boldsymbol{\theta}')$ (an alternative instance) to quantify the hardness of learning the best arm in either environment. In our case, each bandit instance corresponds to an instance of the $\text{MNL}(n, \boldsymbol{\theta})$ problem with the arm set containing all subsets of $[n]$ of size upto $k$: $\mathcal{A} = \{S \subseteq [n] \mid |S| \in [k]\}$. The key of the proof relies on carefully crafting a true instance, with optimal arm $a^* = 1$, and a family of 'slightly perturbed' alternative instances $\{\boldsymbol{\nu}^a : a \neq 1\}$, each with optimal arm $a \neq 1$, chosen as: **(1). True Instance:** $\text{MNL}(n, \boldsymbol{\theta}^1)$ : $\theta_1^1 > \theta_2^1 = \ldots = \theta_n^1 = \theta$ (for some $\theta \in \mathbb{R}_+$), , and for each suboptimal item $a \in [n] \setminus \{1\}$, the **(2). Altered instances:** $\text{MNL}(n, \boldsymbol{\theta}^a)$ : $\theta_a^a = \theta_1^1 + \epsilon = \theta + (\Lambda + \epsilon)$; $\theta_i^a = \theta_i^1$, $\forall i \in [n] \setminus \{a\}$ for some $\epsilon > 0$. The result of Thm. 3 now follows by applying Lemma 13 on pairs of problem instances $(\nu, \nu'^{(a)})$ with suitable upper bounds on the divergences. (Complete proof given in Appendix C.3). □

**Note:** We also show an alternate version of the regret lower bound of $\Omega\left( \dfrac{n}{\left( \underset{i \in [n] \setminus \{a^*\}}{\min} p_{a^*,i} - 0.5 \right)} \ln T \right)$ in terms of pairwise preference-based instance complexities (details are moved to Appendix C.4).

**Improved regret lower bound with Top-$m$-ranking Feedback.** In contrast to the situation with only winner feedback, the following (more general) result shows a reduced lower bound when Top-$m$-ranking Feedback is available in each play of a subset, opening up the possibility of improved learning (regret) performance when ranked-order feedback is available.

**Theorem 4** (Regret Lower Bound: Winner-regret with Top-$m$-ranking Feedback). *For any No-regret algorithm $\mathcal{A}$ for the Winner-regret problem with Top-$m$-ranking Feedback, there exists a problem instance MNL($n, \boldsymbol{\theta}$) such that the expected Winner-regret incurred by $\mathcal{A}$ satisfies* $\underset{T \to \infty}{\lim\inf} \mathbf{E}_{\boldsymbol{\theta}}\left[ \frac{R_T^1(\mathcal{A})}{\ln T} \right] \geq \dfrac{\theta_{a^*}}{\left( \underset{i \in [n] \setminus \{a^*\}}{\min} \frac{\theta_{a^*}}{\theta_i} - 1 \right)} \dfrac{(n-1)}{m}$, *where as in Thm. 3, $\mathbf{E}_{\boldsymbol{\theta}}[\cdot]$ denotes expectation under the algorithm and the MNL model MNL($n, \boldsymbol{\theta}$), and recall $a^* := \arg\max_{i \in [n]} \theta_i$.*

**Proof sketch.** The main observation made here is that the KL divergences for Top-$m$-ranking Feedback are $m$ times compared to the case of Winner Feedback, which we show using chain rule for KL divergences [20]: $KL(p_S^1, p_S^a) = KL(p_S^1(\sigma_1), p_S^a(\sigma_1)) + \sum_{i=2}^{m} KL(p_S^1(\sigma_i \mid \sigma(1 : i-1)), p_S^a(\sigma_i \mid \sigma(1 : i-1)))$, where $\sigma_i = \sigma(i)$ and $KL(P(Y \mid X), Q(Y \mid X)) := \sum_x Pr\left( X = x \right) \left[ KL(P(Y \mid X = x), Q(Y \mid X = x)) \right]$ denotes the conditional KL-divergence. Using this, along

with the upper bound on KL divergences for Winner Feedback (derived for Thm. 3), we show that $KL(p_S^1, p_S^a) \leq \frac{m\Delta_a'^2}{\theta_S^1(\theta_1^1+\epsilon)}$, $\forall a \in [n] \setminus \{1\}$ (where $\theta_S = \sum_{i \in S} \theta_i$ and $\Delta_a' = O(\theta_{a^*} - \theta_a)$), which precisely gives the $\frac{1}{m}$-factor reduction in the lower bound compared to Winner Feedback. The bound now can be derived following a similar technique used for Thm. 3 (details in Appendix C.5). $\square$.

**Remark 4.** Thm. 4 shows a $\Omega\left(\frac{n \ln T}{m}\right)$ lower bound on regret, containing the instance-dependent constant term $\frac{\theta_{a^*}}{\left(\min\limits_{i \in [n] \setminus \{a^*\}} \frac{\theta_{a^*}}{\theta_i} - 1\right)}$ which exposes the hardness of the regret minimisation problem in terms of the 'gap' between the best $a^*$ and the second best item $\min_{i \in [n] \setminus \{a^*\}} \theta_i$: $(\theta_{a^*} - \max_{i \in [n] \setminus \{a^*\}} \theta_j)$. The $\frac{1}{m}$ factor improvement in learning rate with Top-$m$-ranking Feedback can be intuitively interpreted as follows: revealing an $m$-ranking of a $k$-set is worth about $\ln\left(\binom{k}{m}m!\right) = O(m \ln k)$ bits of information, which is about $m$ times as large compared to revealing a single winner.

### 3.2 An order-optimal algorithm for Winner-regret

We here show that above fundamental lower bounds on Winner-regret are, in fact, achievable with carefully designed online learning algorithms. We design an upper-confidence bound (UCB)-based algorithm for Winner-regret with Top-$m$-ranking Feedback model based on the following ideas:

**(1). Playing sets of only** $(m + 1)$ **sizes:** It is enough for the algorithm to play subsets of size either $(m + 1)$ (to fully exploit the Top-$m$-ranking Feedback) or 1 (singleton sets), and not play a singleton unless there is a high degree of confidence about the single item being the best item.

**(2). Parameter estimation from pairwise preferences:** It is possible to play the subset-wise game just by maintaining pairwise preference estimates of all $n$ items of the MNL$(n, \boldsymbol{\theta})$ model using the idea of *Rank-Breaking*–the idea of extracting pairwise comparisons from (partial) rankings and applying estimators on the obtained pairs treating each comparison independently over the received subset-wise feedback—this is possible owning to the independence of irrelevant attributes (IIA) property of the MNL model (Defn. 10).

**(3). A new UCB-based max-min set building rule for playing large sets (*build_S*):** Main novelty of *MaxMin-UCB* lies in its underlying set building subroutine (Alg. 2, Appendix C.1 ), that constructs $S_t$ by applying a recursive max-min strategy on the UCB estimates of empirical pairwise preferences.

**Algorithm description.** *MaxMin-UCB* maintains an pairwise preference matrix $\hat{\mathbf{P}} \in [0, 1]^{n \times n}$, whose $(i, j)$-th entry $\hat{p}_{ij}$ records the empirical probability of $i$ having beaten $j$ in a pairwise duel, and a corresponding upper confidence bound $u_{ij}$ for each pair $(i, j)$. At any round $t$, it plays a subset $S_t \subseteq [n], |S_t| \in [k]$ using the *Max-Min* set building rule *build_S* (see Alg. 2), receives Top-$m$-ranking Feedback $\sigma_t \in \Sigma_{S_t}^m$ from $S_t$, and updates the $\hat{p}_{ij}$ entries of pairs in $S_t$ by applying *Rank-Breaking* (Line 10). The set building rule *build_S* is at the heart of *MaxMin-UCB* which builds the subset $S_t$ from a set of potential Condorcet winners ($\mathcal{C}_t$) of round $t$: By recursively picking the strongest opponents of the already selected items using a max-min selection strategy on $u_{ij}$. The complete algorithm is presented in Alg. 1, Appendix C.1.

The following result establishes that *MaxMin-UCB* enjoys $O(\frac{n}{m} \ln T)$ regret with high probability.

**Theorem 5** (*MaxMin-UCB*: High Probability Regret bound). *Fix a time horizon $T$ and $\delta \in (0, 1)$, $\alpha > \frac{1}{2}$. With probability at least $(1 - \delta)$, the regret of MaxMin-UCB for Winner-regret with Top-$m$-ranking Feedback satisfies* $R_T^1 \leq \left(2\left[\frac{2\alpha n^2}{(2\alpha-1)\delta}\right]^{\frac{1}{2\alpha-1}} + 2D \ln 2D\right)\hat{\Delta}_{\max} + \frac{\ln T}{m+1}\sum_{i=2}^{n}(D_{\max}\hat{\Delta}_i)$, *where $\forall i \in [n] \setminus \{a^*\}$, $\hat{\Delta}_i = (\theta_{a^*} - \theta_i)$, $\Delta_i = \frac{\theta_{a^*} - \theta_i}{2(\theta_{a^*}+\theta_i)}$, $\hat{\Delta}_{\max} = \max_{i \in [n] \setminus \{a^*\}} \hat{\Delta}_i$ $D_{1i} = \frac{4\alpha}{\Delta_i^2}$, $D := \sum_{i<j} D_{ij}$, $D_{\max} = \max_{i \in [n] \setminus \{a^*\}} D_{1i}$.*

**Proof sketch.** The proof hinges on analysing the entire run of *MaxMin-UCB* by breaking it up into 3 phases: (1). **Random-Exploration** (2). **Progress**, and (3). **Saturation**.

**(1). Random-Exploration**: This phase runs from time 1 to $f(\delta) = \left[\frac{2\alpha n^2}{(2\alpha-1)\delta}\right]^{\frac{1}{2\alpha-1}}$, for any $\delta \in (0, 1)$, such that for any $t > f(\delta)$, the upper confidence bounds $u_{ij}$ are guaranteed to be correct for the true values $p_{ij}$ for all pairs $(i, j) \in [n] \times [n]$ (i.e. $p_{ij} \leq u_{ij}$), with high probability $(1 - \delta)$.

**(2). Progress**: After $t > f(\delta)$, the algorithm can be viewed as starting to explore the 'confusing items', appearing in $\mathcal{C}_t$, as potential candidates for the *Best-Item* $a^*$, and trying to capture $a^*$ in a holding set $\mathcal{B}_t$. At any time, the set $\mathcal{B}_t$ is either empty or a singleton by construction, and once

$a^* \in \mathcal{B}_t$ it stays their forever (with high probability). The **Progress** phase ensures that the algorithm explores fast enough so that within a constant number of rounds, $\mathcal{B}_t$ captures $\{a^*\}$.

**(3). Saturation**: This is the last phase from time $T_0(\delta)+1$ to $T$. As the name suggests, *MaxMin-UCB* shows relatively stable behavior here, mostly playing $S_t = \{a^*\}$ and incurring almost no regret.

Although Thm. 5 shows a $(1-\delta)$-high probability regret bound for *MaxMin-UCB* it is important to note that the algorithm itself does not require to take the probability of failure $(\delta)$ as input. As a consequence, by simply integrating the bound obtained in Thm. 5 over the entire range of $\delta \in [0, 1]$, we get an expected regret bound of *MaxMin-UCB* for Winner-regret with Top-$m$-ranking Feedback:

**Theorem 6.** *The expected regret of MaxMin-UCB for Winner-regret with Top-$m$-ranking Feedback is:* $\mathbf{E}[R_T^1] \leq \left( 2 \left[ \frac{2\alpha n^2}{(2\alpha-1)} \right]^{\frac{1}{2\alpha-1}} \frac{2\alpha-1}{\alpha-1} + 2D \ln 2D \right) \hat{\Delta}_{\max} + \frac{\ln T}{m+1} \sum_{i=2}^{n} (D_{\max} \hat{\Delta}_i)$, *in $T$ rounds.*

**Remark 5.** This is an upper bound on expected regret of the same order as that in the lower bound of Thm. 3, which shows that the algorithm is essentially regret-optimal. From Thm. 6, note that the first two terms $\left( 2 \left[ \frac{2\alpha n^2}{(2\alpha-1)} \right]^{\frac{1}{2\alpha-1}} \frac{2\alpha-1}{\alpha-1} + 2D \ln 2D \right) \hat{\Delta}_{\max}$ of $\mathbf{E}[R_T^1]$ are essentially instance specific constants, its only the third term which makes expected regret $O\left( \frac{n \ln T}{m} \right)$ which is in fact optimal in terms of its dependencies on $n$ and $T$ (since it matches the $\Omega\left( \frac{n \ln T}{m} \right)$ lower bound of Thm. 4). Moreover the problem dependent complexity terms $(D_{\max} \hat{\Delta}_i) = \frac{16\alpha(\theta_{a^*}-\theta_i)(\theta_{a^*}+\max_{j\in[n]\setminus\{a^*\}}\theta_j)^2}{(\theta_{a^*}-\max_{j\in[n]\setminus\{a^*\}}\theta_j)^2} \leq \frac{64\alpha(\theta_{a^*}-\theta_i)(\theta_{a^*})}{(\theta_{a^*}-\max_{j\in[n]\setminus\{a^*\}}\theta_j)^2} = O\left( \frac{\theta_{a^*}}{(\theta_{a^*}-\max_{j\in[n]\setminus\{a^*\}}\theta_j)} \right)$, also brings out the inverse dependency on the 'gap-term' $(\theta_{a^*} - \max_{j\in[n]\setminus\{a^*\}}\theta_j)$ as discussed in Rem. 4.

## 4 Minimising Top-$k$-regret

In this section, we study the problem of minimising Top-$k$-regret with Top-$k$-ranking Feedback. As before, we first derive a regret lower bound, for this learning setting, of the form $\Omega\left( \frac{n-k}{k\Delta_{(k)}} \ln T \right)$ (recall $\Delta_{(k)}$ from Sec. 2).We next propose an UCB based algorithm (Alg. 3) for the same, along with a matching upper bound regret analysis (Thm. 8,9) showing optimality of our proposed algorithm.

### 4.1 Regret lower bound for Top-$k$-regret with Top-$k$-ranking Feedback

**Theorem 7** (Regret Lower Bound: Top-$k$-regret with Top-$m$-ranking Feedback). *For any No-regret learning algorithm $\mathcal{A}$ for Top-$k$-regret that uses Top-$k$-ranking Feedback, and for any problem instance MNL($n, \boldsymbol{\theta}$), the expected regret incurred by $\mathcal{A}$ when run on it satisfies* $\liminf_{T \to \infty} \mathbf{E}_{\boldsymbol{\theta}}\left[ \frac{R_T^k(\mathcal{A})}{\ln T} \right] \geq \frac{\theta_1 \theta_{k+1}}{\Delta_{(k)}} \frac{(n-k)}{k}$, *where $\mathbf{E}_{\boldsymbol{\theta}}[\cdot]$ denotes expectation under the algorithm and MNL($n, \boldsymbol{\theta}$) model.*

**Proof sketch.** Similar to 4, the proof again relies on carefully constructing a true instance, with optimal set of *Top-k Best-Items* $S_{(k)} = [k]$, and a family of slightly perturbed alternative instances $\{\boldsymbol{\nu}^a : a \in [n] \setminus S_{(k)}\}$, for each suboptimal arm $a \in [n] \setminus S_{(k)}$, which we design as: **(1). True Instance:** MNL($n, \boldsymbol{\theta}^1$) : $\theta_1^1 = \theta_2^1 = \ldots = \theta_{k-1}^1 = \theta + 2\epsilon; \theta_n^1 = \theta + \epsilon; \theta_{k+1}^1 = \theta_{k+2}^1 = \ldots \theta_{n-1}^1 = \theta$, for some $\theta \in \mathbb{R}_+$ and $\epsilon > 0$. Clearly *Top-k Best-Items* of MNL($n, \boldsymbol{\theta}^1$) is $S_{(k)}[1] = [k-1] \cup \{n\}$. **(2). Altered Instances:** For every $n - k$ suboptimal items $a \notin S_{(k)}[1]$, now consider an altered instance **Instance a,** denoted by MNL($n, \boldsymbol{\theta}^a$), such that $\theta_a^a = \theta + 2\epsilon; \theta_i^a = \theta_i^1, \forall i \in [n] \setminus \{a\}$. The result of Thm. 7 now can be obtained by following an exactly same procedure as described for the proof of Thm. 4. The complete details is given in Appendix D.1. $\square$

**Remark 6.** The regret lower bound of Thm. 7 is $\Omega(\frac{(n-k) \ln T}{k})$, with an instance-dependent term $\frac{\theta_1 \theta_{k+1}}{(\theta_k - \theta_{k+1})}$ which shows for recovering the *Top-k Best-Items*, the problem complexity is governed by the *'gap'* between the $k^{th}$ and $(k+1)^{th}$ best item $\Delta_{(k)} = (\theta_k - \theta_{k+1})$, as consistent with intuition.

### 4.2 An order-optimal algorithm with low Top-$k$-regret with Top-$k$-ranking Feedback

**Main idea: A recursive set-building rule:** As with the *MaxMin-UCB* algorithm (Alg. 1), we maintain pairwise UCB estimates $(u_{ij})$ of empirical pairwise preferences $\hat{p}_{ij}$ via *Rank-Breaking*. However the main difference here lies in the set building rule, as here it is required to play sets of

size exactly $k$. The core idea here is to recursively try to capture the set of *Top-k Best-Items* in an ordered set $\mathcal{B}_t$, and, once the set is assumed to be found with confidence (formally $|\mathcal{B}_t| = k$), to keep playing $\mathcal{B}_t$ unless some other potential good item emerges, which is then played replacing the weakest element ($\mathcal{B}_t(k)$) of $\mathcal{B}_t$. The algorithm is described in Alg. 3, Appendix D.2.

**Theorem 8** (*Rec-MaxMin-UCB*: High Probability Regret bound)**.** *Given a fixed time horizon $T$ and $\delta \in (0,1)$, with high probability $(1 - \delta)$, the regret incurred by Rec-MaxMin-UCB for Top-k-regret admits the bound* $R_T^k \leq \left( 2\left[ \frac{2\alpha n^2}{(2\alpha-1)\delta} \right]^{\frac{1}{2\alpha-1}} + 2\bar{D}^{(k)} \ln\left( 2\bar{D}^{(k)} \right) \right) \Delta'_{\max} +$

$\frac{4\alpha \ln T}{k} \left( \sum_{b=k+1}^{n} \frac{(\theta_k - \theta_b)}{\hat{D}^2} \right)$, *where $D^{(k)}$ is an instance dependent constant (see Lem. 26, Appendix),*

$\Delta'_{\max} = \frac{\left( \sum_{i=1}^{k} \theta_i - \sum_{i=n-k+1}^{n} \theta_i \right)}{k}$, *and $\hat{D} = \min_{g \in [k-1]} (p_{kg} - p_{bg})$.*

**Proof sketch.** Similar to Thm. 5, we prove the above bound dividing the entire run of algorithm *Rec-MaxMin-UCB* into three phases and applying an recursive argument:

**(1). Random-Exploration**: Same as Thm. 5, in this case also this phase runs from time 1 to $f(\delta) = \left[ \frac{2\alpha n^2}{(2\alpha-1)\delta} \right]^{\frac{1}{2\alpha-1}}$, for any $\delta \in (0,1)$, after which, for any $t > f(\delta)$, one can guarantee $p_{ij} \leq u_{ij}$ for all pairs $(i, j) \in [n] \times [n]$, with high probability at least $(1 - \delta)$. (Lem. 15)

**(2). Progress**: The analysis of this phase is quite different from that of Thm. 5: After $t > f(\delta)$, the algorithm starts exploring the items in the set of *Top-k Best-Items* in a *recursive* manner–It first tries to capture (one of) the *Best-Item*s in $\mathcal{B}_t(1)$. Once that slot is secured, it goes on for searching the second *Best-Item* from remaining pool of items and try capturing it in $\mathcal{B}_t(2)$ and so on upto $\mathcal{B}_t(k)$. By definition, the phase ends at, say $t = T_0(\delta)$, when $\mathcal{B}_t = S_{(k)}$. Moreover the update rule of *Rec-MaxMin-UCB* (along with Lem. 15) ensures that $\mathcal{B}_t = S_{(k)} \forall t > T_0(\delta)$. The novelty of our analysis lies in showing that $T_0(\delta)$ is bounded by just a instance dependent complexity term which does not scale with $t$ (Lem. 26), and hence the regret incurred in this phase is also constant.

**(3). Saturation**: In the last phase from time $T_0(\delta) + 1$ to $T$ *Rec-MaxMin-UCB* has already captured $S_{(k)}$ in $\mathcal{B}_t$, and $\mathcal{B}_t = S_{(k)}$ henceforth. Hence the algorithm mostly plays $S_t = S_{(k)}$ without incurring any regret. Only if any item outside $\mathcal{B}_t$ enters into the list of potential *Top-k Best-Items*, it takes a very conservative approach of replacing the 'weakest of $\mathcal{B}_t$ by that element to make sure whether it indeed lies in or outside $S_{(k)}$. However we are able to show that any such suboptimal item $i \notin S_{(k)}$ can not occur for more than $O(\frac{\ln T}{\hat{D}^2})$ times (Lem. 27), combining which over all $[n] \setminus [k]$ suboptimal items finally leads to the desired regret. *The complete details are moved to Appendix D.3.* $\square$

From Theorem 8, we can also derive an expected regret bound for *Rec-MaxMin-UCB* in $T$ rounds is:

**Theorem 9.** *The expected regret incurred by MaxMin-UCB for Top-k-regret is:*

$$\mathbf{E}[R_T^1] \leq \left( 2\left[ \frac{2\alpha n^2}{(2\alpha-1)} \right]^{\frac{1}{2\alpha-1}} \frac{2\alpha-1}{\alpha-1} + 2\bar{D}^{(k)} \ln\left( 2\bar{D}^{(k)} \right) \right) \Delta'_{\max} + \frac{4\alpha \ln T}{k} \left( \sum_{b=k+1}^{n} \frac{(\theta_k - \theta_b)}{\hat{D}^2} \right).$$

**Remark 7.** In Thm. 9, the first two terms $\left( 2\left[ \frac{2\alpha n^2}{(2\alpha-1)\delta} \right]^{\frac{1}{2\alpha-1}} + 2\bar{D}^{(k)} \ln\left( 2\bar{D}^{(k)} \right) \right) \Delta'_{\max}$ of $\mathbf{E}[R_T^k]$ are just some MNL$(n, \boldsymbol{\theta})$ model dependent constants which do not contribute to the learning rate of *Rec-MaxMin-UCB*, and the third term is $O\left( \frac{(n-k) \ln T}{k} \right)$ which varies optimally in terms of on $n, k, T$ matching the $\Omega\left( \frac{(n-k) \ln T}{k} \right)$ lower bound of Thm. 7). Also Rem. 6 indicates an inverse dependency on the 'gap-complexity' $(\theta_k - \theta_{k+1})$, which also shows up in above bound through the component $\frac{(\theta_k - \theta_b)}{\hat{D}^2}$: Let $g^* \in [k-1]$ is the minimizer of $\hat{D}$, then $\frac{(\theta_k - \theta_b)}{\hat{D}^2} = \frac{(\theta_{g^*} + \theta_k)(\theta_b + \theta_{g^*})}{\theta_{g^*}^2 (\theta_k - \theta_b)} \leq \frac{4}{\theta_{g^*}(\theta_k - \theta_{k+1})}$, where the upper bounding follows as $\theta_{g^*} \geq \theta_k > \theta_b$ for any $b \in [n] \setminus [k]$, and $\theta_b \leq \theta_{k+1}$ for any $b$.

## 5 Experiments

In this section we present the empirical evaluations of our proposed algorithm *MaxMin-UCB* (abbreviated as **MM**) on different synthetic datasets, and also compare them with different algorithms. All results are reported as average across 50 runs along with the standard deviations. For this we use 7 different MNL$(n, \boldsymbol{\theta})$ environments as described below:

**MNL**$(n, \boldsymbol{\theta})$ **Environments.** 1. *g1*, 2. *g4*, 3. *arith*, 4. *geo*, 5. *har* all with $n = 16$, and two larger models 6. *arithb*, and 7. *geob* with $n = 50$ items in both. Details are moved to Appendix E.

We compare our proposed methods with the following two baselines which closely applies to our problem setup. Note, as discussed in Sec. 1, none of the existing work exactly addresses our problem. **Algorithms. 1. BD:** The *Battling-Duel* algorithm of [36] with *RUCB* aalgorithm [47] as the dueling bandit blackbox, and **2. Sp-TS:** The *Self-Sparring* algorithm of [39] with Thompson Sampling [1], and **3. MM:** Our proposed method *MaxMin-UCB* for Winner-regret (Alg. 1).

**Comparing Winner-regret with Top-$m$-ranking Feedback (Fig. 1):** We first compare the regret performances for $k = 10$ and $m = 5$. From Fig. 1, it clearly follows that in all cases *MaxMin-UCB* uniformly outperforms the other two algorithms taking the advantage of Top-$m$-ranking Feedback which the other two fail to make use of as they both allow repetitions in the played subsets which can not exploit the rank-ordered feedback to the full extent. Furthermore, the thompson sampling based *Sp-TS* in general exhibits a much higher variance compared to the rest due to its bayesian nature. Also as expected, *g1* and *g4* being comparatively easier instances, i.e. with larger 'gap' $\hat{\Delta}_{\max}$ (see Thm. 3, 4,5, 6 etc. for a formal justification), our algorithm converges much faster on these models.

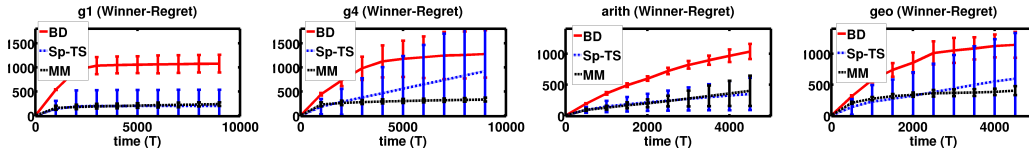

Figure 1: Comparative performances on Winner-regret for $k = 10$, $m = 5$

**Comparing Top-$k$-regret performances for Top-$k$-ranking Feedback (Fig. 2):** We are not aware of any existing algorithm for Top-$k$-regret objective with Top-$k$-ranking Feedback. We thus use a modified version of *Sp-TS* algorithm [39] described above for the purpose–it simply draws $k$-items without repetition and uses *Rank-Breaking* updates to maintain the Beta posteriors. Here again, we see that our method *Rec-MaxMin-UCB* (*Rec-MM*) uniformly outperforms *Sp-TS* in all cases, and as before *Sp-TS* shows a higher variability as well. Interestingly, our algorithm converges the fastest on *g4*, it being the easiest model with largest 'gap' $\Delta_{(k)}$ between the $k^{th}$ and $(k+1)^{th}$ best item (see Thm. 7,8,9 etc.), and takes longest time for *har* since it has the smallest $\Delta_{(k)}$.

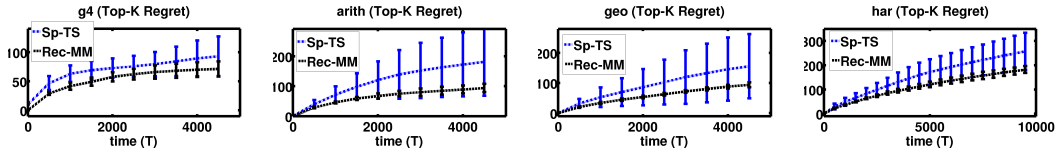

Figure 2: Comparative performances on Top-$k$-regret for $k = 10$

**Effect of varying $m$ with fixed $k$ (Fig. 3):** We also studied our algorithm *MaxMin-UCB*, with varying size rank-ordered feedback $(m)$, keeping the subsetsize $(k)$ fixed, both for Winner-regret and Top-$k$-regret objective, on the larger models *arithb* and *geob* which has $n = 50$ items. As expected, in both cases, regret scales down with increasing $m$ (justifying the bounds in Thm. 5,6),8,9).

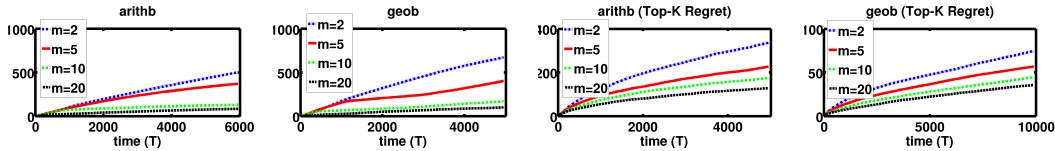

Figure 3: Regret with varying $m$ with fixed $k = 40$ (on our proposed algorithm *MaxMin-UCB*)

## 6 Conclusion and Future Work

Although we have analysed low-regret algorithms for learning with subset-wise preferences, there are several avenues for investigation that open up with these results. The case of learning with contextual subset-wise models is an important and practically relevant problem, as is the problem of considering mixed cardinal and ordinal feedback structures in online learning. Other directions of interest could be studying the budgeted version where there are costs associated with the amount of preference information that may be elicited in each round, or analysing the current problem on a variety of subset choice models, e.g. multinomial probit, Mallows, or even adversarial preference models etc.

**Acknowledgements**

The authors are grateful to the anonymous reviewers for valuable feedback. This work is supported by a Qualcomm Innovation Fellowship 2019, and the Indigenous 5G Test Bed project grant from the Dept. of Telecommunications, Government of India. Aadirupa Saha thanks Arun Rajkumar for the valuable discussions, and the Tata Trusts and ACM-India/IARCS Travel Grants for travel support.

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
