[Supplementary Material · Paper#554-Combinatorial Bandits with Relative Feedback_supplementary.pdf]

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

# Supplementary: Combinatorial Bandits with Relative Feedback

## A  Related Works

Over the last decade, online learning from pairwise preferences has seen a widespread resurgence in the form of the Dueling Bandit problem, from the points of view of both pure-exploration (PAC) settings [44, 40, 13, 12], and regret minimisation [45, 41, 47, 5, 30, 42]. In contrast, bandit learning with combinatorial, subset-wise preferences, though a natural and practical generalisation, has not received a commensurate treatment.

There have been a few attempts in the batch (i.e., non-adaptive) setting for parameter estimation in utility-based subset choice models, e.g. Plackett-Luce or Thurstonian models [23, 18, 29, 26]. In the online setup, a recent work by Brost et al. [11] considers an extension of the dueling bandits framework where multiple arms are chosen in each round, but they receive comparisons for each pair, and there are no regret guarantees stated for their algorithm. Another similar work is DCM-bandits [27], where a list of $k$ distinct items are offered at each round and the users choose one or more from it scanning the list from top to bottom. However due to this cascading nature of their feedback model, this is also not strictly a relative subset-wise preference model unlike ours, since the utility or attraction weight of an item is assumed to be independently drawn, and so their learning objective differs substantially.

A related body of literature lies in dynamic assortment selection, where the goal is to offer a subset of items to customers in order to maximise expected revenue. A specific, bandit (online) counterpart of this problem has been studied in the recent work of Agrawal et al. [2, 3], although it takes items' prices into account due to which their notion of the 'best subset' is rather different from our 'benchmark subset', and the two settings are incomparable in general. More specifically, in this setting,

1. Their assumption of a *no-purchase* option, say item-$0$, necessarily present in every set and having the *known* and *highest* MNL parameter value $\theta_0 = 1$, is crucial for their algorithm design as well as the regret analysis — more specifically this helps them to estimate the MNL model parameters easily. We however do not make this assumption, due to which it is more challenging to estimate the MNL model parameters in our case. This is also precisely the reason why the algorithm of Agrawal et al. [2] cannot be directly applied for solving our problem.

2. The regret objective boils down to the top-$k$ best arm identification problem when all item prices are same, say $r_i = 1, \forall i \in [n]$. So in a sense we actually solve a special case of the assortment selection objective – the top $k$ item(s) – but without assumptions on the no-purchase item with known highest parameter value.

3. Agrawal et al. [2] show gap independent $\tilde{O}(\sqrt{NT})$ regret for their algorithm and this is later improved to gap-dependent $O(n^2 \ln T)$ regret [4]; however, the latter guarantee is suboptimal by a factor of $n$, whereas we show tightness of the regret performance of our proposed algorithms by proving matching lower bound guarantees.

Some recent work addresses the probably approximately correct (PAC) version of the best arm(s) identification problem from subsetwise preferences [17, 35], which is qualitatively different than the optimisation objective considered here. The work which is perhaps closest in spirit to ours is that of Saha and Gopalan [36], but they consider a much more elementary subset choice model based on pairwise preferences, unlike the standard MNL model rooted in choice theory. Sui et al. [39] also address a similar problem; however, a key difference lies in the feedback which consists of outcomes of one or more pairs from the played subset, as opposed to our winner or Top-$m$-ranking Feedback which is often practical.

Lastly, like the dueling bandit, our more general MNL regret problem can be viewed as a stochastic partial monitoring problem [8], in which the reward or loss of a subset play is not directly observed; instead, only stochastic feedback depending on the subset's parameters is observed. Moreover, under one of the regret structures we consider (Winner-regret, Sec. 3.2), playing the optimal subset (the single item with the highest value) yields no useful information.

# B  Properties of MNL model

**Definition 10** (Independence of Irrelevant Alternatives (IIA) property). A choice model is said to possess the *Independence of Irrelevant Attributes (IIA)* property if the ratio of probabilities of choosing any two items, say $i_1$ and $i_2$ from within any choice set $S \ni i_1, i_2$ is independent of a third alternative $j$ present in $S$ [10]. More specifically, $\frac{Pr(i_1|S_1)}{Pr(i_2|S_1)} = \frac{Pr(i_1|S_2)}{Pr(i_2|S_2)}$ for any two distinct subsets $S_1, S_2 \subseteq [n]$ that contain $i_1$ and $i_2$. One such example is the MNL choice model as follows from Defn. 1.

IIA turns out to be very valuable in estimating the parameters of a PL model, with high confidence, via *Rank-Breaking* – the idea of extracting pairwise comparisons from (partial) rankings and applying estimators on the obtained pairs, treating each comparison independently, as described below.

**Definition 11** (**Rank-Breaking** [38, 29]). This is a procedure of deriving pairwise comparisons from multiwise (subsetwise) preference information. Formally, given any set $S \subseteq [n]$, $m \leq |S| < n$, if $\boldsymbol{\sigma} \in \boldsymbol{\Sigma}_S^m$ denotes a possible Top-$m$-ranking Feedback of $S$, *Rank-Breaking* considers each item in $S$ to be beaten by its preceding items in $\boldsymbol{\sigma}$ in a pairwise sense and extracts out total $\sum_{i=1}^m (k-i) = \frac{m(2k-m-1)}{2}$ such pairwise comparisons. For instance, given a full ranking of a set of 4 elements $S = \{a, b, c, d\}$, say $b \succ a \succ c \succ d$, Rank-Breaking generates the set of 6 pairwise comparisons: $\{(b \succ a), (b \succ c), (b \succ d), (a \succ c), (a \succ d), (c \succ d)\}$. Similarly, given the ranking of only 2 most preferred items say $b \succ a$, it yields the 5 pairwise comparisons $(b, a \succ c), (b, a \succ d)$ and $(b \succ a)$ etc. See Line 10 of Algorithm 1 for example.

Owning to the IIA property of MNL$(n, \boldsymbol{\theta})$ model, one can show the following guarantee on the empirical pairwise estimates $\hat{p}_{ij}(T) = \frac{n_i(T)}{n_{ij}(T)}$ obtained via *Rank-Breaking* on MNL based subsetwise preferences:

**Lemma 12** ([37]). *Consider a MNL$(n, \boldsymbol{\theta})$ model, and fix two distinct items $i, j \in [n]$. Let $S_1, \ldots, S_T$ be a sequence of (possibly random) subsets of $[n]$ of size at least 2, where $T$ is a positive integer, and $i_1, \ldots, i_T$ a sequence of random items with each $i_t \in S_t$, $1 \leq t \leq T$, such that for each $1 \leq t \leq T$, (a) $S_t$ depends only on $S_1, \ldots, S_{t-1}$, and (b) $i_t$ is distributed as the Plackett-Luce winner of the subset $S_t$, given $S_1, i_1, \ldots, S_{t-1}, i_{t-1}$ and $S_t$, and (c) $\forall t : \{i, j\} \subseteq S_t$ with probability 1. Let $n_i(T) = \sum_{t=1}^T \mathbf{1}(i_t = i)$ and $n_{ij}(T) = \sum_{t=1}^T \mathbf{1}(\{i_t \in \{i, j\}\})$. Then, for any positive integer $v$, and $\eta \in (0, 1)$,*

$$Pr\left(\frac{n_i(T)}{n_{ij}(T)} - \frac{\theta_i}{\theta_i + \theta_j} \geq \eta, \; n_{ij}(T) \geq v\right) \vee Pr\left(\frac{n_i(T)}{n_{ij}(T)} - \frac{\theta_i}{\theta_i + \theta_j} \leq -\eta, \; n_{ij}(T) \geq v\right) \leq e^{-2v\eta^2}.$$

**Remark 8.** Above lemma is crucially used in proving the regret bounds of our proposed algorithms (Alg. 1 and 3), in particular see the derivation of Lem. 15.

# C  Supplementary for Sec. 3

## C.1  Algorithm Pseudocode for Winner-regret

---
**Algorithm 1** *MaxMin-UCB*
---
1: **init:** $\alpha > 0.5$, $\mathbf{W} \leftarrow [0]_{n \times n}$, $\mathcal{B}_0 \leftarrow \emptyset$
2: **for** $t = 1, 2, 3, \ldots, T$ **do**
3:   Set $\mathbf{N} = \mathbf{W} + \mathbf{W}^\top$, and $\hat{\mathbf{P}} = \frac{\mathbf{W}}{\mathbf{N}}$. Denote $N = [n_{ij}]_{n \times n}$ and $\hat{P} = [\hat{p}_{ij}]_{n \times n}$.
4:   Define $u_{ij} = \hat{p}_{ij} + \sqrt{\frac{\alpha \ln t}{n_{ij}}}$, $\forall i, j \in [n], i \neq j$, $u_{ii} = \frac{1}{2}$, $\forall i \in [n]$. $\mathbf{U} = [u_{ij}]_{n \times n}$
5:   $\mathcal{C}_t \leftarrow \{i \in [n] \mid u_{ij} > \frac{1}{2}, \forall j \in [n] \setminus \{i\}\}$; $\mathcal{B}_t \leftarrow \mathcal{C}_t \cap \mathcal{B}_{t-1}$
6:   **if** $|\mathcal{C}_t| = 1$, **then** set $\mathcal{B}_t \leftarrow \mathcal{C}_t$, $S_t \leftarrow \mathcal{C}_t$, and go to Line 9
7:   **if** $\mathcal{B}_t \neq \emptyset$ **then** set $S_t \leftarrow \mathcal{B}_t$, **else** select any item $a \in \mathcal{C}_t$, and set $S_t \leftarrow \{a\}$
8:   $S_t \leftarrow S_t \cup \textit{build\_S}(\mathbf{U}, S_t, [n] \setminus S_t, m)$
9:   Play $S_t$, and receive: $\boldsymbol{\sigma}_t \in \boldsymbol{\Sigma}_{S_t}^m$
10:   $W(\sigma_t(k'), i) \leftarrow W(\sigma_t(k'), i) + 1$ $\forall i \in S_t \setminus \sigma_t(1 : k')$ for all $k' = 1, 2, \ldots, \min(|S_t| - 1, m)$
11: **end for**
---

**Algorithm 2** *build_S* $(\mathbf{U}, S, I, \ell)$

---

1: **input: U**: UCB matrix of $\hat{\mathbf{P}}$, S: Set to build, I: pool of items $I$, $\ell > 0$: Number of items to draw
2: $\mathcal{C} \leftarrow \{i \in I \mid u_{ij} > \frac{1}{2}, \forall j \in I \setminus \{i\}\}$
3: **while** $|\mathcal{C}| < \ell$ **do**
4: $\quad S \leftarrow S \cup \mathcal{C}; I \leftarrow I \setminus \mathcal{C}; \mathcal{C} \leftarrow \{i \in I \mid u_{ij} > \frac{1}{2}, \forall j \in I \setminus \{i\}\}; \ell \leftarrow \ell - |\mathcal{C}|$
5: **end while**
6: **for** $k' = 2, 3, \ldots, \ell$ **do**
7: $\quad a \leftarrow \underset{c \in I \setminus S}{\arg \max} \left[ \min_{i \in S} u_{ci} \right]; S \leftarrow S \cup \{a\}$
8: **end for**
9: **return:** $S$

---

## C.2 Restating the change of measure Lemma 1 of Kaufmann et al. [28]

**Lemma 13** ([21]). *Given any bandit instance $(A, \boldsymbol{\mu})$, with $A$ being the arm set of MAB, and $\boldsymbol{\mu} = \{\mu_i, \forall i \in A\}$ being the set of reward distributions associated to $A$ with arm 1 having the highest expected reward, for any suboptimal arm $a \in A \setminus \{1\}$, consider an altered bandit instance $\boldsymbol{\mu}^a$ with $a$ being the (unique) optimal arm (the one with highest expected reward) for $\boldsymbol{\mu}^a$, and let $\boldsymbol{\mu}$ and $\boldsymbol{\mu}^a$ be mutually absolutely continuous for all $a \in A \setminus \{1\}$. At any round $t$, let $A_t$ and $Z_t$ denote the arm played and the observation (reward) received, respectively. Let $\mathcal{F}_t = \sigma(A_1, Z_1, \ldots, A_t, Z_t)$ be the sigma algebra generated by the trajectory of a sequential bandit algorithm upto round $t$. Then, for any $\mathcal{F}_T$-measurable random variable $Z$ with values in $[0, 1]$ it satisfies:*

*$\sum_{i \in A} \mathbf{E}_{\boldsymbol{\mu}}[N_i(T)] KL(\mu_i, \mu_i^a) \geq kl(\mathbf{E}_{\boldsymbol{\mu}}[Z], \mathbf{E}_{\boldsymbol{\mu}^a}[Z])$, where $N_i(T)$ denotes the number of pulls of arm $i \in [n]$ in $T$ rounds, $KL$ is the Kullback-Leibler divergence between distributions, and $kl(p, q)$ is the Kullback-Leibler divergence between Bernoulli distributions with parameters $p$ and $q$.*

## C.3 Proof of Thm. 3

**Theorem 3** (Winner-regret Lower Bound). *For any No-regret learning algorithm $\mathcal{A}$ for Winner-regret that uses Winner Feedback, and for any problem instance $MNL(n, \boldsymbol{\theta})$ s.t. $a^* = \underset{i \in [n]}{\arg \max} \theta_i$,*

*the expected regret incurred by $\mathcal{A}$ satisfies $\liminf_{T \to \infty} \mathbf{E}_{\boldsymbol{\theta}} \left[ \frac{R_T^1(\mathcal{A})}{\ln T} \right] \geq \frac{\theta_{a^*}}{\left( \min_{i \in [n] \setminus \{a^*\}} \frac{\theta_{a^*}}{\theta_i} - 1 \right)} (n - 1)$.*

*Proof.* The foundation of the current lower bound analysis stands on the ground on constructing $MNL(n, \boldsymbol{\theta})$ instances, and slightly modified versions of it such that no algorithm can achieve *No-regret* property on these instances without incurring $\Omega(n \ln T)$ regret. We describe the our constructed problem instances below:

Consider an $MNL(n, \boldsymbol{\theta})$ instance with the arm (item) set $A$ containing all subsets of sizes $1, 2, \ldots$ upto $k$ of $[n]$: $A = \{S = (S(1), \ldots S(k')) \subseteq [n] \mid k' \in [k]\}$. Let $MNL(n, \boldsymbol{\theta}^1)$ be the true distribution associated to the bandit arms $[n]$, given by the MNL parameters $\boldsymbol{\theta}^1 = (\theta_1^1, \ldots, \theta_n^1)$, such that $\theta_1^1 > \theta_i^1, \forall i \in [n] \setminus \{1\}$ such that,

**True Instance:** $MNL(n, \boldsymbol{\theta}^1) : \theta_1^1 > \theta_2^1 = \ldots = \theta_n^1 = \theta$ (say).

for some $\theta \in \mathbb{R}_+$. We moreover denote $\Lambda = (\theta_1^1 - \theta)$. Clearly, the *Best-Item* of $MNL(n, \boldsymbol{\theta}^1)$ is $a^* = 1$. Now for every suboptimal item $a \in [n] \setminus \{1\}$, consider the altered problem instance $MNL(n, \boldsymbol{\theta}^a)$ such that:

**Instance a:** $MNL(n, \boldsymbol{\theta}^a) : \theta_a^a = \theta_1^1 + \epsilon = \theta + (\Lambda + \epsilon); \theta_i^a = \theta_i^1, \forall i \in [n] \setminus \{a\}$

for some $\epsilon > 0$. Clearly, the *Best-Item* of $MNL(n, \boldsymbol{\theta}^a)$ is $a^* = a$. Note that, for problem instance $MNL(n, \boldsymbol{\theta}^a) \, a \in [n]$, the probability distribution associated to arm $S \in A$ is given by

$$p_S^a \sim Categorical(p_1, p_2, \ldots, p_k), \text{ where } p_i = Pr(i|S) = \frac{\theta_i^a}{\sum_{j \in S} \theta_j^a}, \; \forall i \in [k], \forall S \in A, \forall a \in [n],$$

since recall that $Pr(i|S)$ is as defined in Defn. 1. Now applying Lem. 13 we get,

$$\sum_{\{S \in A \setminus \{a\} | a \in S\}} \mathbf{E}_{\boldsymbol{\theta}^1}[N_S(T)] KL(p_S^1, p_S^a) \geq kl(\mathbf{E}_{\boldsymbol{\theta}^1}[Z], \mathbf{E}_{\boldsymbol{\theta}^a}[Z]). \tag{1}$$

The above result holds from the straightforward observation that for any arm $S \in \mathcal{A}$ with $a \notin S$, $p_S^1$ is same as $p_S^a$, hence $KL(p_S^1, p_S^a) = 0, \forall S \in A, a \notin S$ or if $S = \{a\}$.

For the notational convenience we will henceforth denote $S^a = \{S \in \mathcal{A} \setminus \{a\} \mid a \in S\}$. Now let us analyse the right hand side of (1), for any set $S \in S^a$. We further denote $\Lambda' = \Lambda + \epsilon = (\theta_1^1 - \theta) + \epsilon$, $k' = |S| \in [k]$, $r = \mathbf{1}(1 \in S)$, $q = (k' - r)$, and $\theta_S^a = \sum_{i \in S} \theta_i^a$ for any $a \in [n]$.

Note that by construction of above problem instances we can further derive that for any $i \in S$:

$$p_S^1(i) = \begin{cases} \frac{r\theta_1^1}{\theta_S^1} = \frac{\theta + \Lambda}{\theta|S| + r\Lambda}, & \text{such that } i = 1, \\ \frac{\theta}{\theta_S^1} = \frac{\theta}{\theta|S| + r\Lambda}, & \text{otherwise.} \end{cases}$$

On the other hand, for problem **Instance-a**, we have that:

$$p_S^a(i) = \begin{cases} \frac{r\theta_1^1}{\theta_S^1 + \Lambda'} = \frac{\theta + \Lambda}{\theta|S| + \Lambda(1+r) + \epsilon}, & \text{such that } i = 1, \\ \frac{\theta_1^1 + \epsilon}{\theta_S^1 + \Lambda'} = \frac{\theta + \Lambda + \epsilon}{\theta|S| + \Lambda(1+r) + \epsilon}, & \text{such that } i = a, \\ \frac{\theta}{\theta_S^1 + \Lambda'} = \frac{\theta}{\theta|S| + \Lambda(1+r) + \epsilon}, & \text{otherwise.} \end{cases}$$

Now using the following upper bound on $KL(\mathbf{p}, \mathbf{q}) \leq \sum_{x \in \mathcal{X}} \frac{p^2(x)}{q(x)} - 1$, $\mathbf{p}$ and $\mathbf{q}$ be two probability mass functions on the discrete random variable $\mathcal{X}$ [33], we get:

$$
\begin{aligned}
KL(p_S^1, p_S^a) &\leq \sum_{i \in S \setminus \{a\}} \left(\frac{\theta_i^1}{\theta_S^1}\right)^2 \left(\frac{\theta_S^a}{\theta_i^a}\right) + \left(\frac{\theta_a^1}{\theta_S^1}\right)^2 \left(\frac{\theta_S^a}{\theta_a^a}\right) - 1 \\
&= \sum_{i \in S \setminus \{a\}} \left(\frac{\theta_i^1}{\theta_S^1}\right)^2 \left(\frac{\theta_S^1 + \Lambda'}{\theta_i^1}\right) + \left(\frac{\theta_a^1}{\theta_S^1}\right)^2 \left(\frac{\theta_S^1 + \Lambda'}{\theta_a^1 + \Lambda'}\right) - 1 \\
&= \left(\frac{\theta_S^1 + \Lambda'}{(\theta_S^1)^2}\right) \left(\sum_{i \in [n] \setminus \{a\}} \theta_i^1 + \frac{(\theta_a^1)^2}{\theta_a^1 + \Lambda'}\right) - 1 \\
&= \left(\frac{\theta_S^1 + \Lambda'}{(\theta_S^1)^2}\right) \left(\frac{\theta_a^1 \theta_S^1 + \Lambda'(\theta_S^1 - \theta_a^1)}{\theta_a^1 + \Lambda'}\right) - 1 \quad \left[\text{replacing} \sum_{i \in [n] \setminus \{a\}} \theta_i^1 = (\theta_S^1 - \theta_a^1)\right] \\
&= \frac{\Lambda'^2(\theta_S^1 - \theta_a^1)}{(\theta_S^1)^2(\theta_a^1 + \Lambda')} \leq \frac{\Lambda'^2}{\theta_S^1(\theta_a^1 + \Lambda')} = \frac{\Lambda'^2}{\theta_S^1(\theta_1^1 + \epsilon)} \\
&= \frac{(\Lambda + \epsilon)^2}{(\theta|S| + r\Lambda)(\theta_1^1 + \epsilon)} \leq \frac{(\Lambda + \epsilon)^2}{\theta|S|(\theta_1^1 + \epsilon)} \tag{2}
\end{aligned}
$$

Let us now analyze the left hand side of (1), with $Z = \frac{N_a(T)}{T}$, where $N_a(T)$ simply denotes the number of times the singleton set containing item $\{a\}$ is played by $\mathcal{A}$, for any suboptimal item $a \in [n] \setminus \{1\}$. Thus we get,

$$kl(\mathbf{E}_{\boldsymbol{\theta}^1}[Z], \mathbf{E}_{\boldsymbol{\theta}^a}[Z]) \geq \left(1 - \frac{\mathbf{E}_{\boldsymbol{\theta}^1}[N_1(T)]}{T}\right) \ln \frac{T}{T - \mathbf{E}_{\boldsymbol{\theta}^a}[N_1(T)]} - \ln 2, \tag{3}$$

where the inequality follows from the fact that for all $(p, q) \in [0, 1]^2$, $kl(p, q) = p \ln \frac{1}{q} + (1 - p) \ln \frac{1}{1-q} + (p \ln p + (1 - p) \ln(1 - p))$, and $p \ln \frac{1}{q} \geq 0$, $(p \ln p + (1 - p) \ln(1 - p)) \geq -\ln 2$.

But now owing to the *No-regret* property (see Defn. 2) of Algorithm $\mathcal{A}$, we have $\mathbf{E}_{\boldsymbol{\theta}^1}[N_a(T)] = o(T^\alpha)$ and $T - \mathbf{E}_{\boldsymbol{\theta}^a}[N_a(T)] = \mathbf{E}_{\boldsymbol{\theta}^a}[\sum_{S \in A, S \neq \{a\}} N_S(T)] = o(T^\alpha)$, $0 < \alpha \leq 1$. Thus from (3), we get

$$\lim_{T \to \infty} \frac{kl(\mathbf{E}_{\boldsymbol{\theta}^1}[Z], \mathbf{E}_{\boldsymbol{\theta}^a}[Z])}{\ln T} \geq \lim_{T \to \infty} \frac{1}{\ln T} \left[ \left( 1 - \frac{\mathbf{E}_{\boldsymbol{\theta}^1}[N_a(T)]}{T} \right) \ln \frac{T}{T - \mathbf{E}_{\boldsymbol{\theta}^a}[N_a(T)]} - \ln 2 \right]$$

$$= \lim_{T \to \infty} \frac{1}{\ln T} \left[ \left( 1 - \frac{o(T^\alpha)}{T} \right) \ln \frac{T}{T^\alpha} - \ln 2 \right] = (1 - \alpha).$$

Combining above with (2) we get:

$$\lim_{T \to \infty} \frac{1}{\ln T} \sum_{\{S \in S^a\}} \mathbf{E}_{\boldsymbol{\theta}^1}[N_S(T)] KL(p_S^1, p_S^a) \geq (1 - \alpha)$$

$$\implies \lim_{T \to \infty} \frac{1}{\ln T} \sum_{\{S \in S^a\}} \mathbf{E}_{\boldsymbol{\theta}^1}[N_S(T)] \frac{\Lambda'^2}{\theta |S| (\theta_1^1 + \epsilon)} \geq (1 - \alpha)$$

$$\implies \lim_{T \to \infty} \frac{1}{\ln T} \sum_{\{S \in S^a\}} \mathbf{E}_{\boldsymbol{\theta}^1}[N_S(T)] \frac{\Lambda'}{|S|} \geq (1 - \alpha) \frac{\theta (\theta_1^1 + \epsilon)}{\Lambda'} \tag{4}$$

Now applying (4) for each modified bandit **Instance-$\boldsymbol{\theta}^a$**, and summing over $(n-1)$ suboptimal items $a \in [n] \setminus \{1\}$ we get,

$$\lim_{T \to \infty} \frac{1}{\ln T} \sum_{a=2}^{n} \sum_{\{S \in S^a\}} \mathbf{E}_{\boldsymbol{\theta}^1}[N_S(T)] \frac{\Lambda'}{|S|} \geq (1 - \alpha)\theta(\theta_1^1 + \epsilon)\frac{(n-1)}{\Lambda'} \tag{5}$$

Now recall that regret of $\mathcal{A}$ on the true instance MNL$(n, \boldsymbol{\theta}^1)$, is given by: $R_T^1(\mathcal{A}) = \sum_{t=1}^{T} \left( \sum_{i \in S_t} \frac{(\theta_1^1 - \theta_i^1)}{|S_t|} \right)$. But this can be equivalently written as:

$$\mathbf{E}_{\boldsymbol{\theta}^1}[R_T^1(\mathcal{A})] = \mathbf{E}_{\boldsymbol{\theta}^1} \left[ \sum_{t=1}^{T} \sum_{i \in S_t} \frac{(\theta_1^1 - \theta_i^1)}{|S_t|} \right]$$

$$= \mathbf{E}_{\boldsymbol{\theta}^1} \left[ \sum_{t=1}^{T} \sum_{S \in A} \mathbf{1}(S_t = S) \sum_{a=2}^{n} \mathbf{1}(a \in S) \frac{(\theta_1^1 - \theta_a^1)}{|S_t|} \right]$$

$$= \mathbf{E}_{\boldsymbol{\theta}^1} \left[ \sum_{a=2}^{n} \sum_{t=1}^{T} \sum_{S \in A} \mathbf{1}(S_t = S) \mathbf{1}(a \in S) \frac{(\theta_1^1 - \theta_a^1)}{|S|} \right]$$

$$= \sum_{a=2}^{n} \sum_{t=1}^{T} \mathbf{E}_{\boldsymbol{\theta}^1} \left[ \sum_{S \in A} \mathbf{1}(S_t = S) \mathbf{1}(a \in S) \frac{(\theta_1^1 - \theta_a^1)}{|S|} \right]$$

$$= \sum_{a=2}^{n} \sum_{S \in A} \mathbf{E}_{\boldsymbol{\theta}^1} \left[ \sum_{t=1}^{T} \mathbf{1}(S_t = S) \mathbf{1}(a \in S) \frac{(\theta_1^1 - \theta_a^1)}{|S|} \right]$$

$$= \sum_{a=2}^{n} \sum_{S \in A} \left[ \mathbf{E}_{\boldsymbol{\theta}^1}[N_S(T)] \mathbf{1}(a \in S) \frac{(\theta_1^1 - \theta)}{|S|} \right]$$

$$= \sum_{a=2}^{n} \sum_{\{S \in A | a \in S\}} \mathbf{E}_{\boldsymbol{\theta}^1}[N_S(T)] \frac{\Lambda}{|S|} \tag{6}$$

Then combining (6) with (5) we get and taking $\epsilon \to 0$:

$$\lim_{T\to\infty} \frac{1}{\ln T} \mathbf{E}_{\boldsymbol{\theta}^1}[R_T^1(\mathcal{A})] \geq \lim_{T\to\infty} \frac{1}{\ln T} \sum_{a=2}^{n} \sum_{\{S\in S^a\}} \mathbf{E}_{\boldsymbol{\theta}^1}[N_S(T)] \frac{\Lambda}{|S|}$$

$$\geq (1-\alpha)\theta(\theta_1^1)\frac{(n-1)}{\Lambda} = (1-\alpha)\theta_1^1 \frac{(n-1)}{(\frac{\theta_1^1}{\theta}-1)}.$$

Finally, since $\alpha$ is a fixed constant in $(0,1]$, above construction shows the existence of a MNL$(n, \boldsymbol{\theta})$ problem instance, precisely MNL$(n, \boldsymbol{\theta}^1)$, such that for large $T$, $\mathbf{E}_{\boldsymbol{\theta}^1}[R_T^1] = \Omega\left(\frac{\theta_1^1}{\left(\frac{\theta_1^1}{\theta}-1\right)}(n-1)\ln T\right)$, which concludes the proof. $\qquad\square$

### C.4 An alternate version of the regret lower bound (Thm. 4) with pairwise preference-based instance complexities

**Theorem 14** (Alternate version of Thm. 4 with pairwise preference based instance complexities). *For any No-regret algorithm $\mathcal{A}$ for Winner-regret with Winner Feedback, there exists a problem instance of MNL$(n, \boldsymbol{\theta})$ model, such that the expected regret incurred by $\mathcal{A}$ on it satisfies $\liminf\limits_{T\to\infty} \mathbf{E}_{\boldsymbol{\theta}}\left[\frac{R_T^1(\mathcal{A})}{\ln T}\right] \geq \frac{\theta_{a^*}}{4\left(\min\limits_{i\in[n]\setminus\{a^*\}} p_{a^*,i} - 0.5\right)}(n-1)$, where $p_{ij} := Pr(i|\{i,j\}) = \frac{\theta_i}{\theta_i+\theta_j} \;\forall i,j \in [n]$, and $\mathbf{E}_{\boldsymbol{\theta}}[\cdot]$, $a^*$ are same as that of Thm. 3. Thus the only difference lies in terms of the instance dependent complexity term ('gap') which is now expressed in terms of pairwise preference of the best item $a^*$ over the second best item: $\left(\min\limits_{i\in[n]\setminus\{a^*\}} p_{a^*,i} - 0.5\right)$.*

*Proof.* Firstly, is easy to note that $\arg\min\limits_{i\in[n]\setminus\{a^*\}}\left(p_{a^*,i} - 0.5\right) = \arg\max\limits_{i\in[n]\setminus\{a^*\}} \theta_i =: b$ (say). The proof now follows from the fact that

$$p_{a^*b} - 0.5 = \frac{\theta_{a^*}-\theta_b}{2(\theta_{a^*}+\theta_b)} \leq \frac{\theta_{a^*}-\theta_b}{4\theta_b} \quad (\text{since}\,\theta_b \leq \theta^{a^*})$$

Thus using the lower bound from Thm. 4, one can further derive

$$\liminf_{T\to\infty} \frac{1}{\ln T}\mathbf{E}_{\boldsymbol{\theta}}\left[R_T^1(\mathcal{A})\right] \geq \frac{4\theta_{a^*}\theta_b}{4\left(\theta_{a^*}-\theta_b\right)}(n-1) \geq \frac{\theta_{a^*}}{4\left(\min\limits_{i\in[n]\setminus\{a^*\}} p_{a^*,i} - 0.5\right)}(n-1),$$

which proves the claim. $\qquad\square$

### C.5 Proof of Thm. 4

**Theorem 4** (Regret Lower Bound: Winner-regret with Top-$m$-ranking Feedback). *For any No-regret algorithm $\mathcal{A}$ for the Winner-regret problem with Top-$m$-ranking Feedback, there exists a problem instance MNL$(n, \boldsymbol{\theta})$ such that the expected Winner-regret incurred by $\mathcal{A}$ satisfies $\liminf\limits_{T\to\infty} \mathbf{E}_{\boldsymbol{\theta}}\left[\frac{R_T^1(\mathcal{A})}{\ln T}\right] \geq \frac{\theta_{a^*}}{\left(\min\limits_{i\in[n]\setminus\{a^*\}} \frac{\theta_{a^*}}{\theta_i}-1\right)} \frac{(n-1)}{m}$, where as in Thm. 3, $\mathbf{E}_{\boldsymbol{\theta}}[\cdot]$ denotes expectation under the algorithm and the MNL model MNL$(n, \boldsymbol{\theta})$, and recall $a^* := \arg\max_{i\in[n]} \theta_i$.*

*Proof.* The proof proceeds almost same as the proof of Thm. 3, the only difference lies in the analysis of the KL-divergence terms with Top-$m$-ranking Feedback.

Consider the exact same MNL$(n, \boldsymbol{\theta})$ instances, MNL$(n, \boldsymbol{\theta}^a)$ we constructed for Thm. 3. It is now interesting to note that how Top-$m$-ranking Feedback affects the KL-divergence analysis, precisely the KL-divergence shoots up by a factor of $m$ which in fact triggers an $\frac{1}{m}$ reduction in regret learning rate. Note that for Top-$m$-ranking Feedback for any problem instance MNL$(n, \boldsymbol{\theta}^a)$, $a \in [n]$, each

$k$-set $S \subseteq [n]$ (such that $|S| = k$) is associated to $\binom{k}{m}(m!)$ number of possible outcomes, each representing one possible ranking of set of $m$ items of $S$, say $S_m$. Also the probability of any permutation $\boldsymbol{\sigma} \in \Sigma_{S_m}$ is given by $p_S^a(\boldsymbol{\sigma}) = Pr_{\boldsymbol{\theta}^a}(\boldsymbol{\sigma}|S)$, where $Pr_{\boldsymbol{\theta}^a}(\boldsymbol{\sigma}|S)$ is as defined for Top-$m$-ranking Feedback (in Sec. 2.1). For ease of analysis let us first assume $1 \notin S$ and let $k' = |S|$ be the cardinality of $S$ and $m' = min(m,k)$. (Note if $m' \le m+1$ the corresponding Top-$m$-ranking Feedback becomes a full ranking feedback on the entire $m'$ items). In this case we get

$$p_S^1(\boldsymbol{\sigma}) = \prod_{i=1}^{m'} \frac{\theta_{\sigma(i)}^1}{\sum_{j=i}^{m'} \theta_{\sigma(j)}^1 + \sum_{j \in S \setminus \sigma(1:m')} \theta_{\sigma(j)}^1} = \frac{1}{k'(k'-1)(k'-2)\cdots(k'-m+1)}, \quad \forall \sigma \in \Sigma_S^{m'}.$$

On the other hand, for problem **Instance-a**, we have that:

$$p_S^a(\boldsymbol{\sigma}) = \prod_{i=1}^{m'} \frac{\theta_{\sigma(i)}^a}{\sum_{j=i}^{m'} \theta_{\sigma(j)}^a + \sum_{j \in S \setminus \sigma(1:m')} \theta_{\sigma(j)}^a}$$

$$= \begin{cases} \frac{x}{(x+k'-1)(x+k'-2)\cdots(x+k'-i)(k'-i)(k'-i-1)\cdots(k'-m'+1)}, & \text{such that } \sigma(i) = a, \\ \frac{1}{(x+k'-1)(x+k'-2)\cdots(x+k'-i)(k'-i)(k'-i-1)\cdots(k'-m'+1)}, & \text{such that } a \notin \sigma(1:m'), \end{cases}$$

where we denote by $x = 1 + \frac{\Lambda'}{\theta}$, where recall that we denote $\Lambda' = \Lambda + \epsilon$. Similarly we can derive the probability distribution associated to sets including item 1.

The important thing now to note is that $KL(p_S^1, p_S^a) = 0$ for any set $S \not\ni a$. Hence while comparing the KL-divergence of instances $\boldsymbol{\theta}^1$ vs $\boldsymbol{\theta}^a$, we need to focus only on sets containing $a$. Applying *Chain-Rule* of KL-divergence, we now get

$$KL(p_S^1, p_S^a) = KL(p_S^1(\sigma_1), p_S^a(\sigma_1)) + KL(p_S^1(\sigma_2 \mid \sigma_1), p_S^a(\sigma_2 \mid \sigma_1)) + \cdots$$
$$+ KL(p_S^1(\sigma_m \mid \sigma(1:m-1)), p_S^a(\sigma_m \mid \sigma(1:m-1))), \tag{7}$$

where we abbreviate $\sigma(i)$ as $\sigma_i$ and following the usual convention the notation $KL(P(Y \mid X), Q(Y \mid X)) := \sum_x Pr\Big(X = x\Big)\Big[KL(P(Y \mid X = x), Q(Y \mid X = x))\Big]$ denotes the conditional KL-divergence. Moreover it is easy to note that for any $\sigma \in \Sigma_S^m$ such that $\sigma(i) = a$, we have $KL(p_S^1(\sigma_{i+1} \mid \sigma(1:i)), p_S^a(\sigma_{i+1} \mid \sigma(1:i))) := 0$, for all $i \in [m]$.

Now as derived in (2) in the proof of Thm. 3, we have

$$KL(p_S^1(\sigma_1), p_S^a(\sigma_1)) \le \frac{(\Lambda + \epsilon)^2}{\theta |S| (\theta_1^1 + \epsilon)}.$$

To bound the remaining terms of (7), note that for all $i \in [m-1]$

$$KL(p_S^1(\sigma_{i+1} \mid \sigma(1:i)), p_S^a(\sigma_{i+1} \mid \sigma(1:i)))$$
$$= \sum_{\sigma' \in \Sigma_S^i} Pr(\sigma') KL(p_S^1(\sigma_{i+1} \mid \sigma(1:i)) = \sigma', p_S^a(\sigma_{i+1} \mid \sigma(1:i)) = \sigma')$$
$$= \sum_{\sigma' \in \Sigma_S^i \mid a \notin \sigma'} \left[ \prod_{j=1}^{i} \left( \frac{\theta_{\sigma_j'}^1}{\theta_S^1 - \sum_{j'=1}^{j-1} \theta_{\sigma_{j'}'}} \right) \right] \frac{\Lambda'^2}{(|S|-i)\theta(\theta_1^1 + \epsilon)}$$
$$= \prod_{j=1}^{i} (|S|-j) \frac{\theta^i}{\prod_{j=1}^{i}(\theta(|S|-i+1) + \Lambda')} \frac{(\Lambda')^2}{(|S|-i)\theta(\theta_1^1 + \epsilon)} = \frac{\not\theta}{(\theta|S| + \Lambda')} \frac{\Lambda'^2}{\not\theta(\theta_1^1 + \epsilon)}$$
$$= \frac{\Lambda'^2}{(\theta|S| + \Lambda')(\theta_1^1 + \epsilon)},$$

where for simplicity we assumed $1 \notin S$. It is easy to note that the similar analysis would lead to the same upper bound for sets $S$ containing 1 as well. Thus applying above in (7) we get:

$$
\begin{aligned}
KL(p_S^1, p_S^a) &= KL(p_S^1(\sigma_1) + \cdots + KL(p_S^1(\sigma_m \mid \sigma(1:m-1)), p_S^a(\sigma_m \mid \sigma(1:m-1)))) \\
&\leq \frac{m\Lambda'^2}{|S|\theta(\theta_1^1 + \epsilon)}.
\end{aligned} \tag{8}
$$

Eqn. (8) gives the main result to derive Thm. 4 as it shows an $m$-factor blow up in the KL-divergence terms owning to Top-$m$-ranking Feedback. The rest of the proof follows exactly the same argument used in 3. We add the steps below for convenience. Firstly, considering $Z = \frac{N_a(T)}{T}$, in this case as well, one can show that:

$$
\begin{aligned}
\lim_{T \to \infty} \frac{kl(\mathbf{E}_{\boldsymbol{\theta}^1}[Z], \mathbf{E}_{\boldsymbol{\theta}^a}[Z])}{\ln T} &\geq \lim_{T \to \infty} \frac{1}{\ln T}\left[\left(1 - \frac{\mathbf{E}_{\boldsymbol{\theta}^1}[N_a(T)]}{T}\right)\ln\frac{T}{T - \mathbf{E}_{\boldsymbol{\theta}^a}[N_a(T)]} - \ln 2\right] \\
&= \lim_{T \to \infty} \frac{1}{\ln T}\left[\left(1 - \frac{o(T^\alpha)}{T}\right)\ln\frac{T}{T^\alpha} - \ln 2\right] = (1 - \alpha).
\end{aligned}
$$

Now combining above with (8) we get:

$$
\begin{aligned}
&\lim_{T \to \infty} \frac{1}{\ln T} \sum_{\{S \in S^a\}} \mathbf{E}_{\boldsymbol{\theta}^1}[N_S(T)]KL(p_S^1, p_S^a) \geq (1 - \alpha) \\
&\implies \lim_{T \to \infty} \frac{1}{\ln T} \sum_{\{S \in S^a\}} \mathbf{E}_{\boldsymbol{\theta}^1}[N_S(T)]\frac{m\Lambda'^2}{\theta|S|(\theta_1^1 + \epsilon)} \geq (1 - \alpha) \\
&\implies \lim_{T \to \infty} \frac{1}{\ln T} \sum_{\{S \in S^a\}} \mathbf{E}_{\boldsymbol{\theta}^1}[N_S(T)]\frac{\Lambda'}{|S|} \geq (1 - \alpha)\frac{\theta(\theta_1^1 + \epsilon)}{m\Lambda'}
\end{aligned} \tag{9}
$$

Applying (9) for each modified bandit **Instance-$\boldsymbol{\theta}^a$**, and summing over $(n-1)$ suboptimal items $a \in [n] \setminus \{1\}$ we get,

$$
\lim_{T \to \infty} \frac{1}{\ln T} \sum_{a=2}^{n} \sum_{\{S \in S^a\}} \mathbf{E}_{\boldsymbol{\theta}^1}[N_S(T)]\frac{\Lambda'}{|S|} \geq (1 - \alpha)\theta(\theta_1^1 + \epsilon)\frac{(n-1)}{m\Lambda'} \tag{10}
$$

Further recall that we derived earlier that $\mathbf{E}_{\boldsymbol{\theta}^1}[R_T^1(\mathcal{A})] = \sum_{a=2}^{n} \sum_{\{S \in A | a \in S\}} \mathbf{E}_{\boldsymbol{\theta}^1}[N_S(T)]\frac{\Lambda}{|S|}$, using which combined with (10), and taking $\epsilon \to 0$ we get:

$$
\begin{aligned}
\lim_{T \to \infty} \frac{1}{\ln T}\mathbf{E}_{\boldsymbol{\theta}^1}[R_T^1(\mathcal{A})] &\geq \lim_{T \to \infty} \frac{1}{\ln T} \sum_{a=2}^{n} \sum_{\{S \in S^a\}} \mathbf{E}_{\boldsymbol{\theta}^1}[N_S(T)]\frac{\Lambda}{|S|} \\
&\geq (1 - \alpha)\theta(\theta_1^1)\frac{(n-1)}{m\Lambda} = (1 - \alpha)\theta_1^1\frac{(n-1)}{m(\frac{\theta_1^1}{\theta} - 1)}.
\end{aligned}
$$

Now since $\alpha$ is a fixed constant in $(0, 1]$, we thus prove the existence of a MNL$(n, \boldsymbol{\theta})$ problem instance $\big($precisely MNL$(n, \boldsymbol{\theta}^1)\big)$, such that for large $T$, $\mathbf{E}_{\boldsymbol{\theta}^1}[R_T^1] = \Omega\left(\frac{\theta_1^1}{\left(\frac{\theta_1^1}{\theta} - 1\right)}\frac{(n-1)}{m}\ln T\right)$, which concludes the proof.

$\square$

## C.6 Proof of Thm. 5

**Theorem 5** (*MaxMin-UCB*: High Probability Regret bound). *Fix a time horizon $T$ and $\delta \in (0,1)$, $\alpha > \frac{1}{2}$. With probability at least $(1-\delta)$, the regret of MaxMin-UCB for Winner-regret with Top-$m$-ranking Feedback satisfies* $R_T^1 \leq \left( 2\left[ \frac{2\alpha n^2}{(2\alpha-1)\delta} \right]^{\frac{1}{2\alpha-1}} + 2D\ln 2D \right)\hat{\Delta}_{\max} + \frac{\ln T}{m+1}\sum_{i=2}^{n}(D_{\max}\hat{\Delta}_i)$, *where* $\forall i \in [n] \setminus \{a^*\}$, $\hat{\Delta}_i = (\theta_{a^*} - \theta_i)$, $\Delta_i = \frac{\theta_{a^*} - \theta_i}{2(\theta_{a^*}+\theta_i)}$, $\hat{\Delta}_{\max} = \max_{i \in [n]\setminus\{a^*\}} \hat{\Delta}_i$ $D_{1i} = \frac{4\alpha}{\Delta_i^2}$, $D := \sum_{i<j} D_{ij}$, $D_{\max} = \max_{i \in [n]\setminus\{a^*\}} D_{1i}$.

*Proof.* For the notational convenience we will assume $\theta_1 > \theta_2 \dots \geq \theta_n$, so $a^* = 1$. We also use $\hat{p}_{ij}(t), n_{ij}(t)$ and $u_{ij}(t)$ to denote the values of the respective quantities at time iteration $t$, for any $t \in [T]$, just to be precise

$$u_{ij}(t) = \hat{p}_{ij}(t) + \sqrt{\frac{\alpha\ln t}{n_{ij}(t)}}, \forall i,j \in [n], i \neq j,$$

and $u_{ii}(t) = \frac{1}{2}$ for all $i \in [n]$. We also find it convenient to denote

$$c_{ij}(t) = \sqrt{\frac{\alpha\ln t}{n_{ij}(t)}}, \text{ and } l_{ij}(t) = 1 - u_{ij}(t).$$

We also denote $T_0(\delta) = 2f(\delta) + 2D\ln 2D$, where $D := \sum_{i<j} D_{ij}$. We start with the following crucial lemma that analyzes the confidence bounds $[l_{ij}(t), u_{ij}(t)]$ on the pairwise probability estimates $\hat{p}_{ij}$ for each pair $(i,j)$, $i \neq j$.

**Lemma 15.** *Suppose* $\mathbf{P} := [p_{ij}]$ *be the pairwise probability matrix associated to the underlying* $MNL(n, \boldsymbol{\theta})$ *model, i.e.* $p_{ij} = Pr(i|\{i,j\}) = \frac{\theta_i}{\theta_i + \theta_j}$. *Then for any* $\alpha > \frac{1}{2}$, $\delta \in (0,1)$,

$$Pr\left( \forall t > f(\delta), \forall i,j, \ p_{ij} \in [l_{ij}(t), u_{ij}(t)] \right) > (1-\delta)$$

*where* $f(\delta) = \left[ \frac{2\alpha n^2}{(2\alpha-1)\delta} \right]^{\frac{1}{2\alpha-1}}$.

*Proof.* The proof of this lemma is adapted from a similar result (Lemma 1) of [47]. Suppose $\mathcal{G}_{ij}(t)$ denotes the event that at time $t$, $p_{ij} \in [l_{ij}(t), u_{ij}(t)]$, $\forall i,j \in [n]$. $\mathcal{G}_{ij}^c(t)$ denotes its complement.

**Case 1:** $(i = j)$ Note that for any such that pair $(i,i)$, $\mathcal{G}_{ii}(t)$ always holds true for any $t \in [T]$ and $i \in [n]$, as $p_{ii} = u_{ii} = l_{ii} = \frac{1}{2}$.

**Case 2:** $(i \neq j)$ Recall from the definition of $u_{ij}(t)$ that $\mathcal{G}_{ij}(t)$ equivalently implies at round $t$, $|\hat{p}_{ij}(t) - p_{ij}| \leq \sqrt{\frac{\alpha\ln(t)}{n_{ij}(t)}}$, $\forall i,j \in [n]$. Moreover, for any $t$ and $i,j$, $\mathcal{G}_{ij}(t)$ holds if and only if $\mathcal{G}_{ij}(t)$ as $|\hat{p}_{ji}(t) - p_{ji}| = |(1 - \hat{p}_{ij}(t)) - (1 - p_{ij})| = |\hat{p}_{ij}(t) - p_{ij}|$. Thus we will restrict our focus only to pairs $i < j$ for the rest of the proof.

Let $\tau_{ij}(n)$ the time step $t \in [T]$ when the pair $(i,j)$ was updated for the $n^{th}$ time. Clearly for any $n \in \mathbb{N}$, $\tau_{ij}(n+1) \geq \tau_{ij}(n)$ and $\tau_{ij}(n+k) > \tau_{ij}(n)$. For convenience of notation we use $F = f(\delta)$. It is now straightforward to note that we want to find $F$ such that:

$$Pr\left( \forall t > F, \forall i,j \text{ such that } i < j, \ \mathcal{G}_{ij}(t) \right) > (1-\delta) \text{ or equivalently,}$$
$$Pr\left( \exists t > F \text{ and atleast a pair } (i < j), \text{ with } \mathcal{G}_{ij}^c(t) \right) < \delta. \tag{11}$$

Further decomposing the right hand side of above we get:

$$Pr\left(\exists t > F, i < j, \text{ such that } \mathcal{G}_{ij}^c(t)\right)$$

$$\leq \sum_{i<j}\left[Pr\left(\exists n \geq 0, \tau_{ij}(n) > F, |p_{ij} - \hat{p}_{ij}(\tau_{ij}(n))| > \sqrt{\frac{\alpha \ln(\tau_{ij}(n))}{n_{ij}(\tau_{ij}(n))}}\right)\right]$$

$$\leq \sum_{i<j}\left[Pr\left(\exists n \leq F, \tau_{ij}(n) > F, |p_{ij} - \hat{p}_{ij}(n)| > \sqrt{\frac{\alpha \ln(\tau_{ij}(n))}{n}}\right) + Pr\left(\exists n > F, |p_{ij} - \hat{p}_{ij}(n)| > \sqrt{\frac{\alpha \ln(\tau_{ij}(n))}{n}}\right)\right],$$

where $\hat{p}(n) = \frac{w_{ij}(\tau_{ij}(n))}{w_{ij}(\tau_{ij}(n)) + w_{ij}(\tau_{ji}(n))}$ is the frequentist estimate of $p_{ij}$ after $n$ comparisons between arm $i$ and $j$. Now the above inequality can be further upper bounded as:

$$Pr\left(\exists t > F, i < j, \text{ such that } \mathcal{G}_{ij}^c(t)\right)$$

$$\leq \sum_{i<j}\left[Pr\left(\exists n \leq F, \tau_{ij}(n) > F, |p_{ij} - \hat{p}_{ij}(n)| > \sqrt{\frac{\alpha \ln(F)}{n}}\right) + Pr\left(\exists n > F, |p_{ij} - \hat{p}_{ij}(n)| > \sqrt{\frac{\alpha \ln(n)}{n}}\right)\right],$$

since in the second term $\tau_{ij}(n) > F$, and for the third term $n < \tau_{ij}(n)$ since at a particular time iteration, any pair $(i, j)$, can be updated at most once, implying $n \leq \tau_{ij}(n)$. Using Lem. 12 we now get:

$$Pr\left(\exists t > F, i < j, \text{ such that } \mathcal{G}_{ij}^c(t)\right)$$

$$\leq \sum_{i<j}\left[\sum_{n=1}^{F} 2e^{-2n\frac{\alpha \ln F}{n}} + \sum_{n=F+1}^{\infty} 2e^{-2n\frac{\alpha \ln n}{n}}\right]$$

$$= \frac{n(n-1)}{2}\left[2\sum_{n=1}^{F}\frac{1}{F^{2\alpha}} + \sum_{n=F+1}^{\infty}\frac{2}{n^{2\alpha}}\right]$$

$$\leq \frac{n^2}{F^{2\alpha-1}} + n^2\int_{F}^{\infty}\frac{dx}{x^{2\alpha}} \leq \frac{n^2}{F^{2\alpha-1}} - \frac{n^2}{(1-2\alpha)F^{2\alpha-1}} = \frac{(2\alpha)n^2}{(2\alpha-1)F^{2\alpha-1}}.$$

Now from (11), we want to find $F$ such that

$$\frac{2\alpha n^2}{(2\alpha-1)F^{2\alpha-1}} \leq \delta,$$

which suffices by setting $F = \left[\frac{2\alpha n^2}{(2\alpha-1)\delta}\right]^{\frac{1}{2\alpha-1}}$, and recall that we assumed $f(\delta) = F$, for any given $\delta \in [0, 1]$, which concludes the claim. $\qquad\square$

Lem. 15 ensures the termination of the **Random-Exploration** phase. We now proceed to analyse **Progress** phase which shows that the set $\mathcal{B}_t$ captures the *Best-Item* $a^* = 1$ 'soon after' $f(\delta)$ within a constant number of rounds $T_0(\delta)$ which is independent of $T$ (see Lem. 20). Once the 1 is captured in $\mathcal{B}_t$, the algorithm goes into **Saturation** phase where the suboptimal items can not stay too long in the set of potential *Best-Item*s $\mathcal{C}_t$, and thus the regret bound follows (Lem. 22). More formally, the rest of the proof follows based on the following main observations:

**In Progress:**

- **Observation** 1: At any iteration, the set $\mathcal{B}_t$ is either singleton or an empty set.

- **Observation** 2: For any $\delta \in (0, 1)$, suppose $T_0(\delta) := \min_{t > f(\delta)} \mathcal{B}_t = \{1\}$, then for any $t > T_0(\delta)$, $\mathcal{B}_t = \{1\}$.

- **Observation** 3: $T_0(\delta)$ is not far from $f(\delta)$ (see Lem. 20 which holds due to Lem. 18 and 19)

**In Saturation:**

- **Observation** 4**:** After $T_0(\delta)$, $\mathcal{B}_t = \{1\}$ thereafter, and thus it is always played in $S_t$, i.e. $1 \in S_t$ for all $t > T_0(\delta)$. Now the suboptimal items start getting frequently compared to item 1 every time they are played alongside with 1, and thus they can not stay too long in the set of 'good' items $\mathcal{C}_t$ and eventually $\mathcal{C}_t = \{1\}$, when the algorithm *MaxMin-UCB* plays the optimal set $S_t = \{1\}$ only, and thus the regret bound follows. (see Lem. 21 and 22)

Observation 1 is straightforward to follow from Alg. 1. Observation 2 follows from Lem. 15, as for any $t > f(\delta)$, $1 \in \mathcal{C}_t$ always, since $u_{1i} \geq p_{1i} > \frac{1}{2}$, $\forall i \in [n] \setminus \{1\}$. We next recall the notations before proceeding to the next results: Let $\Delta_i = P(1 \succ i) - \frac{1}{2} = \frac{\theta_1 - \theta_i}{2(\theta_1 + \theta_i)}$, for all $i \in [n] \setminus \{1\}$. For any pair $(i,j)$ such that $1 \notin \{i,j\}$, we define $D_{ij} = \frac{4\alpha}{\min\{\Delta_i^2, \Delta_j^2\}}$. For any $i \in [n] \setminus \{1\}$, $D_{1i} = \frac{4\alpha}{\Delta_i^2}$.

**Definition 16** (Unsaturated Pairs). At any time $t \in [T]$, and any pair of two distinct items $i, j \in [n]$, we term the pair $(i,j)$ to be *unsaturated* at time $t$ if $n_{ij}(t) \leq D_{ij} \ln t$. Otherwise, we call the pair *saturated* at $t$.

**Lemma 17.** *For any set $S \subseteq [n]$ such that $|S| \geq m + 1$, and given a Top-$m$-ranking Feedback $\sigma \in \Sigma_S^m$ (for any $m \in [k-1]$), applying pairwise Rank-Breaking on $S$ according to $\sigma$, updates each element $i \in S$ for atleast $m$ distinct pairs.*

*Proof.* For any item $i \in S$, one can make the following two case analyses:

**Case 1:** ($i \in \sigma(1:m)$). If the item $i$ occurs in one of the top-$m$ position, it is clearly compared with rest of the $|S| - 1 \geq m$ elements of $S$, as it is beaten by the preceding items in $\sigma$ and wins over the rest.

**Case 2:** ($i \notin \sigma(1:m)$). In this case $i$ gets updated for $m$ many pairs since it is considered to be beaten by all items in $\sigma(1:m)$ in a pairwise duel.

The claim follows combining Case 1 and 2 above. $\qquad\square$

**Lemma 18.** *Assuming $\forall t > f(\delta)$, and $\forall i, j \in [n]$ $p_{ij} \in [l_{ij}(t), u_{ij}(t)]$, for some $\delta \in (0, 1)$: At any iteration $t > f(\delta)$, if $\exists$ a suboptimal item $i \in [n] \setminus \{1\}$, such that $i \in \mathcal{C}_t$, then the pair $(1, i)$ is unsaturated at $t$.*

*Proof.* Firstly note that for any $t > f(\delta)$, $1 \in \mathcal{C}_t$ always, since $u_{1i} \geq p_{1i} > \frac{1}{2}$, $\forall i \in [n] \setminus \{1\}$.

Now suppose $(1, i)$ is indeed saturated at time $t$, i.e. $n_{1i}(t) > D_{1i} \ln t$, then this implies:

$$u_{i1}(t) = \hat{p}_{i1}(t) + c_{i1}(t) \leq p_{i1}(t) + 2c_{i1}(t) = p_{i1}(t) + \Delta_i = \frac{1}{2},$$

which implies $i \notin \mathcal{C}_t$, at $t$. Thus $(1, i)$ must be unsaturated at $t$. $\qquad\square$

**Lemma 19.** *Assuming $\forall t > f(\delta)$, and $\forall i, j \in [n]$ $p_{ij} \in [l_{ij}(t), u_{ij}(t)]$, for some $\delta \in (0, 1)$: At any iteration $t > f(\delta)$, for any set $S \not\ni 1$ if $\exists$ a suboptimal item $a \in [n] \setminus \{1\}$, such that $a = \arg\max_{c \in I \setminus S} \left[ \min_{i \in S} u_{ci}(t) \right]$, then $\exists$ atleast one suboptimal item $i \in S$ such that the pair $(i, a)$ is unsaturated at $t$.*

*Proof.* We start by noting that for any $i \in \mathcal{C}_t \setminus \{1\}$ if $u_{ji}(t) > u_{1i}(t)$, then $n_{ij} \leq D_{ij} \ln t$, i.e. the pair $(i, j)$ must be unsaturated at round $t$. Suppose not and $n_{ij}(t) > D_{ij} \ln t$. Then we have that

$$u_{ij}(t) - l_{ij}(t) = 2c_{ij}(t) \leq \sqrt{\min\{\Delta_i^2, \Delta_j^2\}} = \min\{\Delta_i, \Delta_j\}.$$

But on the other hand, since $u_{ji}(t) > u_{1i}(t)$, this implies:

$$u_{ij}(t) - l_{ij}(t) = u_{ij}(t) + u_{ji}(t) - 1 > \frac{1}{2} + u_{1i}(t) - 1 > \frac{1}{2} + p_{1i}(t) - 1 = \Delta_i \geq \min\{\Delta_i, \Delta_j\},$$

where the first inequality is because $i \in \mathcal{C}_t$, hence $u_{ij}(t) > \frac{1}{2}$ and $u_{ji}(t) > u_{1i}(t)$. This leads to a contradiction implying that $(i, j)$ has to be unsaturated at $t$.

The proof now follows noting that, by definition of $a$, $\min_{i \in S} u_{ai}(t) > \min_{i \in S} u_{1i}(t) \implies \exists$ atleast one item $i \in S$ such that $u_{ai}(t) > u_{1i}$. But following above chain of argument that leads to a contradiction unless the pair $(i, a)$ is unsaturated at round $t$. $\qquad \square$

Combining Lem. 18 and 19 we can conclude that it does not take too long to reach to a time $T_0(\delta) > f(\delta)$, such that $\mathcal{C}_{T_0(\delta)} = \{1\}$ and thus $\mathcal{B}_t = \{1\}$ for all $t > T_0(\delta)$.

**Lemma 20.** *Assume $\forall t > f(\delta)$, and $\forall i, j \in [n]$ $p_{ij} \in [l_{ij}(t), u_{ij}(t)]$, for some $\delta \in (0, 1)$. Then if we define $T_0(\delta)$ such that: $T_0(\delta) = \min\{t > f(\delta) \mid \mathcal{C}_t = \{1\}\}$, it can be upper bounded as $T_0(\delta) \le 2f(\delta) + 2D \ln 2D$, where $D := \sum_{i<j} D_{ij}$.*

*Proof.* The first observation for this is to note that: For any $t > f(\delta)$, $1 \in \mathcal{C}_t$ since $u_{1i} \ge p_{1i} > \frac{1}{2}$, $\forall i \in [n] \setminus \{1\}$. So, until $T_0(\delta)$, for all $t \in \{f(\delta), f(\delta) + 1, \ldots T_0(\delta) - 1\}$, $|\mathcal{C}_t| \ge 2$.

Secondly, for any $t \in \{f(\delta), f(\delta) + 1, \ldots T_0(\delta) - 1\}$, there exists atleast $\min(m, |\mathcal{C}_t| - 1)$ unsaturated pairs in $S_t$ which gets updated. This holds from the following two case analyses:

**Case 1:** ($1 \in S_t$). This is the easy case since for any item $i \in \mathcal{C}_t \setminus \{1\}$, we know that $(1, i)$ is unsaturated from Lem. 18, and item 1 has to be updated for atleast $\min(m, |\mathcal{C}_t| - 1)$ many unsaturated pairs as follows from Lem. 17.

**Case 2:** ($1 \notin S_t$). From Lem. 19 we know that for any item $i \in S_t \cap \mathcal{C}_t \setminus \{1\}$ has to be unsaturated with atleast another item $j \in S_t$. Since *MaxMin-UCB* makes sure $|S_t| \ge m + 1$, again owing to Lem. 17, any item $i \in S_t \cap \mathcal{C}_t \setminus \{1\}$ gets compared for atleast $m$ pairs out of which atleast one pair has to be unsaturated which proves the claim.

Moreover, as argued above, at any round $t \in \{f(\delta), f(\delta) + 1, \ldots T_0(\delta) - 1\}$, since $|\mathcal{C}_t| \ge 2$, any such round $t$ updates atleast $\min(m, |\mathcal{C}_t| - 1) \ge 1$ unsaturated pair.

Thirdly, at any time $t$, if all pairs $(i, j)$, $i \ne j$, $i, j \in [n]$ are saturated, then $\mathcal{C}_t = \{1\}$.

So to bound $T_0(\delta)$, all we need to figure out is the worst possible number of iterations *MaxMin-UCB* would take to saturate all possible unsaturated pairs, precisely $\sum_{i<j} D_{ij} \ln t$ many pairwise updates. But as we argued before, since any round $t > f(\delta)$ updates atleast one unsaturated pair, we find that

$$T_0(\delta) = \min\{t > f(\delta) \mid t > f(\delta) + \sum_{i<j} D_{ij} \ln t\}$$

Now it is easy to see that the above inequality ($t > f(\delta) + \sum_{i<j} D_{ij} \ln t$) certainly satisfies for $t = 2f(\delta) + 2D \ln 2D$, where $D := \sum_{i<j} D_{ij}$ as:

$$
\begin{aligned}
f(\delta) + D \ln t &= f(\delta) + D \ln(2f(\delta) + 2D \ln 2D) \\
&\le f(\delta) + D \ln(2D \ln 2D) + D \frac{2f(\delta)}{2D \ln 2D} \\
&\le f(\delta) + D \ln(2D)^2 + f(\delta), \quad [\text{ since, } \ln 2D > 1] \\
&= 2f(\delta) + 2D \ln 2D = t.
\end{aligned}
$$

Since $T_0(\delta)$ is the minimum time index at which $t > f(\delta) + D \ln t$ is satisfied, clearly $T_0(\delta) \le 2f(\delta) + 2D \ln 2D$. $\qquad \square$

Finally we are ready to prove Thm. 5 based on the the following two claims:

**Lemma 21.** *Assume $\forall t > f(\delta)$, and $\forall i, j \in [n]$ $p_{ij} \in [l_{ij}(t), u_{ij}(t)]$, for some $\delta \in (0, 1)$. For any time step $t > T_0(\delta)$, $1 \in S_t$ always. Moreover for any $|S_t| > 1$, item 1 gets compared with atleast $m$ suboptimal items $a \in [n] \setminus \{1\}$.*

*Proof.* For any $t > f(\delta)$, $1 \in \mathcal{C}_t$ since $u_{1i} \geq p_{1i} > \frac{1}{2}$, $\forall i \in [n] \setminus \{1\}$. Moreover as $T_0(\delta)$ ensures $\mathcal{C}_{T_0(\delta)} = \{1\}$, at this round, the algorithm set $\mathcal{B}_{T_0(\delta)} = \{1\}$. For the subsequent rounds $t > T_0(\delta)$, thus the algorithm continues setting $\mathcal{B}_t = \mathcal{B}_{t-1} \cap \mathcal{C}_t = \{1\}$.

Moreover note that for any $t > T_0(\delta)$, unless $|\mathcal{C}_t| = 1$, the algorithm always plays a set $S_t$ such that $|S_t| = m + 1$, and in which item 1 always resides. Then by Lem. 17 we can conclude that item 1 is compared with atleast $m$ distinct items at any round after pairwise *Rank-Breaking* update. $\square$

**Lemma 22.** *For any $\delta \in (0, 1)$, with probability atleast $(1 - \delta)$, the total cumulative regret of MaxMin-UCB is upper bounded as:*

$$R_T^1 \leq \left( 2 \Big[ \frac{2\alpha n^2}{(2\alpha - 1)\delta} \Big]^{\frac{1}{2\alpha - 1}} + 2D \ln 2D \right) \hat{\Delta}_{\max} + \frac{\ln T}{m+1} \sum_{i=2}^{n} \hat{\Delta}_i \big( \mathbf{1}(m = 1) D_{1i} + \mathbf{1}(m > 1) D_{\max} \big)$$

*where recall that $\forall i \in [n] \setminus \{a^*\}$, $\hat{\Delta}_i = (\theta_1 - \theta_i)$, $\Delta_i = p_{1i} = \frac{\theta_1 - \theta_i}{2(\theta_1 + \theta_i)}$, $\hat{\Delta}_{\max} = \max_{i \in [n] \setminus \{1\}} \hat{\Delta}_i$*
*$D_{1i} = \frac{4\alpha}{\Delta_i^2}$, $D := \sum_{i<j} D_{ij}$, $D_{\max} = \max_{i \in [n] \setminus \{1\}} D_{1i}$.*

*Proof.* Given Lem. 21 in place, the crucial observation now is to note that for any $t > T_0(\delta)$, *MaxMin-UCB*, always explores as long as there exists any suboptimal item $i \in [n] \setminus \{1\}$ such that the pair $(1, i)$ is unsaturated and thus $i \in \mathcal{C}_t$. In other words, our set building rule (*build_S*) always picks items from $\mathcal{C}_t$ first before picking anything from $[n] \setminus \mathcal{C}_t$. However, any suboptimal item $i \in [n] \setminus \{1\}$ can belong to $\mathcal{C}_t$ only if the pair $(1, i)$ is unsaturated, as follows from Lem. 18.

Thus for any time $t$, if the pair $(1, i)$ is already saturated (i.e. $n_{1i}(t) > D_{1i}(t) \ln t$), then $i \notin S_t$ unless item 1 is saturated with every suboptimal item in $[n] \setminus \{1\}$. But then $\mathcal{C}_t = \{1\}$ by Lem. 18 and the algorithm would go on playing $S_t = \{1\}$ until some pair $(1, i)$ gets unsaturated again. This argument holds true even for $t = T$.

Now lets try to analyse what is the maximum number of time an item $i \in [n] \setminus \{1\}$ can show up at any round post **Saturation** (i.e. for any $t > T_0(\delta)$). But since post **Saturation**, for any $t > T_0(\delta)$, $1 \in S_t$ always, the quantity $n_{1i}(T) - n_{1i}(T_0(\delta))$ is same as above. We hence analyse $n_{1i}(T) - n_{1i}(T_0(\delta))$ with the following two cases:

**Case-1** $(m = 1)$: This case is easy to analyse since at any round $t$, $|S_t| = 2$ and since $1 \in S_t$, so 1 gets compared with exactly one other suboptimal element $i \in [n] \setminus \{1\}$ at any $t$ such that $i \in \mathcal{C}_t$. So clearly $n_{1i}(T) - n_{1i}(T_0(\delta)) \leq D_{1i} \ln T$ as by Lem. 18 after 1 is compared to $i$ for $D_{1i} \ln T$ times $i \notin \mathcal{C}_t$ henceforth.

**Case-2** $(m > 1)$: In this case there are two possible ways $i$ can show up in $S_t$: $(i)$. If its unsaturated with 1 for which it can show up for at most $D_{1i} \ln T$ times as argued i Case-1, and $(ii)$. When $i \notin \mathcal{C}_t$ but it shows up as a place holder for onle of the $m + 1$ slot of $S_t$ as long as some other element $j \in [n] \setminus \{1\}$, $i \neq j$ is unsaturated with 1 and $j \in \mathcal{C}_t$. But in the worst case once all item $i \in [n] \setminus \{1\}$ has appeared in $S_t$ for $D_{\max} \ln T (> D_{1i} \ln T)$ times by Lem. 1 we have $u_{i1} < \frac{1}{2} \forall i \in [n] \setminus \{1\}$ and then $\mathcal{C}_t$ has to be the singleton $\{1\}$ thereafter. So it has to be that $n_{1i}(T) - n_{1i}(T_0(\delta)) \leq D_{\max} \ln T$.

Finally note that all our above results holds good under the assumption that $\forall t > f(\delta)$, and $\forall i, j \in [n]$ $p_{ij} \in [l_{ij}(t), u_{ij}(t)]$, for some $\delta \in (0, 1)$, which itself holds good with probability atleast $(1 - \delta)$. Thus we have the maximum regret incurred by *MaxMin-UCB* in $T$ rounds is

$$R_T \leq T_0(\delta)\hat{\Delta}_{\max} + \frac{1}{m+1} \sum_{i=2}^{n} \big( n_{1i}(T) - n_{1i}(T_0(\delta)) \big) \big( \mathbf{1}(m = 1) D_{1i} + \mathbf{1}(m > 1) D_{\max} \big)$$

$$= \left( 2 \Big[ \frac{2\alpha n^2}{(2\alpha - 1)\delta} \Big]^{\frac{1}{2\alpha - 1}} + 2D \ln 2D \right) \hat{\Delta}_{\max} + \frac{\ln T}{m+1} \sum_{i=2}^{n} \hat{\Delta}_i \big( \mathbf{1}(m = 1) D_{1i} + \mathbf{1}(m > 1) D_{\max} \big),$$

with probability atleast $(1 - \delta)$, were first term in the right hand side of the inequality holds since the maximum possible per trial regret that could be incurred by *MaxMin-UCB* in initial $T_0(\delta)$ rounds is $\hat{\Delta}_{\max}$. The proof now follows further upper bounding $T_0$ using Lem. 20. $\square$

This also concludes the proof of Thm. 5 using the exact value of $f(\delta)$ as derived in Lem. 15.

$\square$

## C.7 Proof of Theorem 6

**Theorem 6.** *The expected regret of MaxMin-UCB for Winner-regret with Top-$m$-ranking Feedback is:* $\mathbf{E}[R_T^1] \leq \left( 2\left[ \frac{2\alpha n^2}{(2\alpha-1)} \right]^{\frac{1}{2\alpha-1}} \frac{2\alpha-1}{\alpha-1} + 2D\ln 2D \right) \hat{\Delta}_{\max} + \frac{\ln T}{m+1} \sum_{i=2}^{n} (D_{\max}\hat{\Delta}_i)$*, in $T$ rounds.*

*Proof.* Recall from the statement of Thm. 5 that the only term in $R_T^1$ that depends on $\delta$ is $2f(\delta)$, where recall that $T_0(\delta) = 2f(\delta) + 2D\ln 2D$. Then by integrating $f(\delta)$ for $\delta$ from 0 to 1 as follows:

$$\int_0^1 f(\delta)d\delta = \int_0^1 \left[ \frac{2\alpha n^2}{(2\alpha-1)\delta} \right]^{\frac{1}{2\alpha-1}} d\delta = \left[ \frac{2\alpha n^2}{(2\alpha-1)} \right]^{\frac{1}{2\alpha-1}} \int_0^1 \left( \frac{1}{\delta} \right)^{\frac{1}{2\alpha-1}} d\delta = \left[ \frac{2\alpha n^2}{(2\alpha-1)} \right]^{\frac{1}{2\alpha-1}} \frac{2\alpha-1}{2\alpha-2}$$

Thus expected regret $\mathbf{E}_\delta[R_T]$ can be upper bounded as:

$$\mathbf{E}_\delta[R_T^1] \leq \left( 2\left[ \frac{2\alpha n^2}{(2\alpha-1)} \right]^{\frac{1}{2\alpha-1}} \frac{2\alpha-1}{\alpha-1} + 2D\ln 2D \right) \hat{\Delta}_{\max} + \frac{\ln T}{m+1} \sum_{i=2}^{n} \hat{\Delta}_i \left( \mathbf{1}(m=1)D_{1i} + \mathbf{1}(m>1)D_{\max} \right).$$

$\square$

# D Supplementary for Section 4

## D.1 Proof of Thm. 7

**Theorem 7** (Regret Lower Bound: Top-$k$-regret with Top-$m$-ranking Feedback). *For any No-regret learning algorithm $\mathcal{A}$ for Top-$k$-regret that uses Top-$k$-ranking Feedback, and for any problem instance MNL$(n, \boldsymbol{\theta})$, the expected regret incurred by $\mathcal{A}$ when run on it satisfies $\liminf_{T\to\infty} \mathbf{E}_{\boldsymbol{\theta}} \left[ \frac{R_T^k(\mathcal{A})}{\ln T} \right] \geq \frac{\theta_1 \theta_{k+1}}{\Delta_{(k)}} \frac{(n-k)}{k}$, where $\mathbf{E}_{\boldsymbol{\theta}}[\cdot]$ denotes expectation under the algorithm and MNL$(n, \boldsymbol{\theta})$ model.*

*Proof.* The main idea lies in constructing 'hard enough' problem instances for which any *No-regret* algorithm has to incur $\Omega\left( \frac{n}{k\Delta_{(k)}} \ln T \right)$ regret.

We choose our *true problem instance* with MNL parameters $\boldsymbol{\theta}^1 = (\theta_1^1, \ldots, \theta_n^1)$, such that:

**True Instance:** MNL$(n, \boldsymbol{\theta}^1)$ : $\theta_1^1 = \theta_2^1 = \ldots = \theta_{k-1}^1 = \theta + 2\epsilon$;
$$\theta_n^1 = \theta + \epsilon; \theta_{k+1}^1 = \theta_{k+2}^1 = \ldots \theta_{n-1}^1 = \theta.$$

for some $\theta \in \mathbb{R}_+$ and $\epsilon > 0$. Clearly, the *Top-$k$ Best-Items* (recall the definition from Def. 1, Sec. 2) of MNL$(n, \boldsymbol{\theta}^1)$ is $S_{(k)}[1] = [k-1] \cup \{n\}$. Now for every $n-k$ suboptimal items $a \notin S_{(k)}[1]$, consider the altered problem instance MNL$(n, \boldsymbol{\theta}^a)$ such that:

**Instance a:** MNL$(n, \boldsymbol{\theta}^a)$ : $\theta_a^a = \theta + 2\epsilon$; $\theta_i^a = \theta_i^1, \ \forall i \in [n] \setminus \{a\}$

And now the *Top-$k$ Best-Items* of MNL$(n, \boldsymbol{\theta}^a)$ is $S_{(k)}[a] = [k-1] \cup \{a\}$. Same as the case for proof of Thm. 3 or Thm. 4, we now again use the results of [21] (Lem. 13) for proving the lower bound. Precisely, the main trick lies in analyzing the KL-divergence terms for the above problem instances. For ease of analysis we first assume analyse the case with just the Winner Feedback. Borrowing same notations from Thm. 4, and denoting $x = |S \cap S_{(k)}[1]| - r, r = \mathbf{1}(n \in S), y = k - (x + r)$, for any set $S \in S^a$, we now get that for any $i \in S$:

$$p^1_S(i) = \begin{cases} \frac{\theta+2\epsilon}{\theta^1_S} = \frac{\theta+2\epsilon}{x(\theta+2\epsilon)+r(\theta+\epsilon)+y\theta} = \frac{\theta+2\epsilon}{k\theta+\epsilon(2x+r)}, & \text{such that } i \in S_{(k)}[1] \cap S, \\ \frac{\theta+\epsilon}{\theta^1_S} = \frac{\theta+\epsilon}{x(\theta+2\epsilon)+r(\theta+\epsilon)+y\theta} = \frac{\theta+\epsilon}{k\theta+\epsilon(2x+r)}, & \text{such that } i = n, \\ \frac{\theta}{\theta^1_S} = \frac{\theta}{x(\theta+2\epsilon)+r(\theta+\epsilon)+y\theta} = \frac{\theta+\epsilon}{k\theta+\epsilon(2x+r)}, & \text{otherwise.} \end{cases}$$

On the other hand, for problem **Instance-a**, we have that:

$$p^a_S(i) = \begin{cases} \frac{\theta+2\epsilon}{\theta^a_S} = \frac{\theta+2\epsilon}{(x+1)(\theta+2\epsilon)+r(\theta+\epsilon)+(y-1)\theta} = \frac{\theta+2\epsilon}{k\theta+\epsilon(2(x+1)+r)}, & \text{such that } i \in (S \cap S_{(k)}[1]) \cup \{a\}, \\ \frac{\theta+\epsilon}{\theta^a_S} = \frac{\theta+\epsilon}{(x+1)(\theta+2\epsilon)+r(\theta+\epsilon)+(y-1)\theta} = \frac{\theta+\epsilon}{k\theta+\epsilon(2(x+1)+r)}, & \text{such that } i = n, \\ \frac{\theta}{\theta^a_S} = \frac{\theta}{(x+1)(\theta+2\epsilon)+r(\theta+\epsilon)+(y-1)\theta} = \frac{\theta+\epsilon}{k\theta+\epsilon(2(x+1)+r)}, & \text{otherwise.} \end{cases}$$

For ease of notation we denote $\theta_S = \theta^1_S$. Now using the following upper bound on $KL(\mathbf{p},\mathbf{q}) \leq \sum_{x \in \mathcal{X}} \frac{p^2(x)}{q(x)} - 1$, $\mathbf{p}$ and $\mathbf{q}$ be two probability mass functions on the discrete random variable $\mathcal{X}$ [33], we get for any $S \in S^a$:

$$KL(p^1_S, p^a_S) \leq \sum_{i \in S} \left(\frac{\theta^1_i}{\theta^1_S}\right)^2 \left(\frac{\theta^a_S}{\theta^a_i}\right) - 1$$

$$= \sum_{i \in S \cap S_{(k)}[1]} \left(\frac{\theta+2\epsilon}{\theta_S}\right)^2 \left(\frac{\theta_S+2\epsilon}{\theta+2\epsilon}\right)$$

$$+ \sum_{i \in S \cap (\{a,n\} \cup S_{(k)}[1])^c} \left(\frac{\theta}{\theta_S}\right)^2 \left(\frac{\theta_S+2\epsilon}{\theta}\right) + \left(\frac{\theta+\epsilon}{\theta_S}\right)^2 \left(\frac{\theta_S+2\epsilon}{\theta+\epsilon}\right) + \left(\frac{\theta}{\theta_S}\right)^2 \left(\frac{\theta_S+2\epsilon}{\theta+2\epsilon}\right) - 1$$

$$= \frac{\theta+2\epsilon}{\theta_S^2} \left[\theta_S + \frac{\theta^2}{\theta+2\epsilon} - \theta\right] = \frac{2\epsilon}{\theta_S}\left[1 - \frac{\theta}{\theta+2\epsilon}\right] - \frac{4\epsilon^2\theta}{\theta_S^2(\theta+2\epsilon)}$$

$$\leq \frac{(2\epsilon)^2[\theta_S - \theta]}{\theta_S^2(\theta+2\epsilon)} \leq \frac{(2\epsilon)^2}{\theta_S(\theta+2\epsilon)} = \frac{4\epsilon^2}{[k\theta+(2x+r)\epsilon](\theta+2\epsilon)} \leq \frac{4\epsilon^2}{k\theta(\theta+2\epsilon)} \tag{12}$$

Now coming back to the Top-$k$-ranking Feedback applying chain rule of KL-divergence (similar to the analysis of Eqn. (7)), we can write

$$KL(p^1_S, p^a_S) = KL(p^1_S(\sigma_1) + \cdots + KL(p^1_S(\sigma_k \mid \sigma(1:k-1)), p^a_S(\sigma_k \mid \sigma(1:k-1)))).$$

for any ranking $\sigma \in \Sigma^k_S$. And following the same argument that of (8), we further get

$$KL(p^1_S, p^a_S) \leq \frac{4k\epsilon^2}{k\theta(\theta+2\epsilon)}. \tag{13}$$

The rest of the proof follows exactly the same argument used in 4. We add the steps below for convenience. As before, considering $Z = \frac{N_{S_{(k)}[1]}(T)}{T}$, for large $T$, in this case we get:

$$\lim_{T \to \infty} \frac{kl(\mathbf{E}_{\boldsymbol{\theta}^1}[Z], \mathbf{E}_{\boldsymbol{\theta}^a}[Z])}{\ln T} \geq (1-\alpha), \tag{14}$$

which follows from an exact similar analysis shown in the proof of Thm. 3 along with the facts that:

$$\mathbf{E}_{\boldsymbol{\theta}^1}[N_{S_{(k)}[1]}(T)] = 1 - o(T^\alpha) \quad \text{since } \mathcal{A} \text{ is assumed to be } \textit{No-regret} \text{ and}$$
$$\mathbf{E}_{\boldsymbol{\theta}^a}[N_{S_{(k)}[1]}(T)] = o(T^\alpha).$$

Then using the results of Eqn. (13) and (14) in Lem. 13, we further get:

$$\lim_{T\to\infty} \frac{1}{\ln T} \sum_{\{S\in S^a\}} \mathbf{E}_{\boldsymbol{\theta}^1}[N_S(T)] KL(p_S^1, p_S^a) \geq (1-\alpha)$$

$$\implies \lim_{T\to\infty} \frac{1}{\ln T} \sum_{\{S\in S^a\}} \mathbf{E}_{\boldsymbol{\theta}^1}[N_S(T)] \frac{4k\epsilon^2}{k\theta(\theta+2\epsilon)} \geq (1-\alpha)$$

$$\implies \lim_{T\to\infty} \frac{1}{\ln T} \sum_{\{S\in S^a\}} \mathbf{E}_{\boldsymbol{\theta}^1}[N_S(T)] \frac{\epsilon}{k} \geq (1-\alpha)\frac{\theta(\theta+2\epsilon)}{4k\epsilon} \tag{15}$$

Now applying (15) for each $n-k$ modified bandit **Instance-$\boldsymbol{\theta}^a$** (i.e. for each $a \in [n] \setminus S_{(k)}[1]$), we get:

$$\lim_{T\to\infty} \frac{1}{\ln T} \sum_{a=k+1}^{n} \sum_{\{S\in S^a\}} \mathbf{E}_{\boldsymbol{\theta}^1}[N_S(T)] \frac{\epsilon}{k} \geq (1-\alpha)\theta(\theta+2\epsilon)\frac{(n-k)}{4k\epsilon} \tag{16}$$

Further recall from Eqn. (2.2) the expected regret of $\mathcal{A}$ on problem instance MNL$(n, \boldsymbol{\theta}^1)$ is given by:
$\mathbf{E}_{\boldsymbol{\theta}^1}[R_T^k(\mathcal{A})] = \sum_{t=1}^{T} r_t^k = \sum_{t=1}^{T} \left( \frac{\sum_{i\in[k]} \theta_i^1 - \sum_{i\in S_t} \theta_i^1}{k} \right)$ which can be rewritten as:

$$\mathbf{E}_{\boldsymbol{\theta}^1}[R_T^k(\mathcal{A})] = \mathbf{E}_{\boldsymbol{\theta}^1}\left[ \sum_{t=1}^{T} \left( \frac{\sum_{i\in[k]} \theta_i^1 - \sum_{i\in S_t} \theta_i^1}{k} \right) \right]$$

$$\geq \mathbf{E}_{\boldsymbol{\theta}^1}\left[ \sum_{t=1}^{T} \left( \frac{\sum_{i\in[k]} \theta_k^1 - \sum_{i\in S_t} \theta_i}{k} \right) \right]$$

$$= \mathbf{E}_{\boldsymbol{\theta}^1}\left[ \sum_{t=1}^{T} \left( \sum_{i\in[S_t]} \frac{\theta_k^1 - \theta_i}{k} \right) \right]$$

$$= \mathbf{E}_{\boldsymbol{\theta}^1}\left[ \sum_{t=1}^{T} \sum_{S\in A} \mathbf{1}(S_t = S) \sum_{a=k+1}^{n} \mathbf{1}(a \in S) \frac{(\theta_k^1 - \theta_a^1)}{|S_t|} \right]$$

$$= \mathbf{E}_{\boldsymbol{\theta}^1}\left[ \sum_{a=k+1}^{n} \sum_{t=1}^{T} \sum_{S\in A} \mathbf{1}(S_t = S)\mathbf{1}(a \in S) \frac{((\theta+\epsilon)-\theta)}{k} \right] \quad (\text{since } \theta_k^1 = \theta_n^1 = \theta+\epsilon)$$

$$= \sum_{a=k+1}^{n} \sum_{t=1}^{T} \mathbf{E}_{\boldsymbol{\theta}^1}\left[ \sum_{S\in A} \mathbf{1}(S_t = S)\mathbf{1}(a \in S) \frac{\epsilon}{k} \right]$$

$$= \sum_{a=k+1}^{n} \sum_{S\in A} \mathbf{E}_{\boldsymbol{\theta}^1}\left[ \sum_{t=1}^{T} \mathbf{1}(S_t = S)\mathbf{1}(a \in S) \frac{\epsilon}{k} \right]$$

$$= \sum_{a=k+1}^{n} \sum_{S\in A} \left[ \mathbf{E}_{\boldsymbol{\theta}^1}[N_S(T)]\mathbf{1}(a \in S) \frac{\epsilon}{k} \right]$$

$$= \sum_{a=k+1}^{n} \sum_{\{S\in A|a\in S\}} \mathbf{E}_{\boldsymbol{\theta}^1}[N_S(T)] \frac{\epsilon}{k} \tag{17}$$

Using above combined with (16) we get:

$$\lim_{T\to\infty}\frac{1}{\ln T}\mathbf{E}_{\boldsymbol{\theta}^1}[R_T^k(\mathcal{A})] \geq \lim_{T\to\infty}\frac{1}{\ln T}\sum_{a=k+1}^{n}\sum_{\{S\in A|a\in S\}}\mathbf{E}_{\boldsymbol{\theta}^1}[N_S(T)]\frac{\epsilon}{k}$$

$$\geq \lim_{T\to\infty}\frac{1}{\ln T}\sum_{a=k+1}^{n}\sum_{\{S\in S^a\}}\mathbf{E}_{\boldsymbol{\theta}^1}[N_S(T)]\frac{\epsilon}{k} \geq (1-\alpha)\theta(\theta+2\epsilon)\frac{(n-k)}{4k\epsilon}.$$

Now since $\alpha$ is a fixed constant in $(0,1]$, we thus prove the existence of a MNL$(n,\boldsymbol{\theta})$ problem instance $\big($precisely MNL$(n,\boldsymbol{\theta}^1)\big)$, such that for large $T$, $\mathbf{E}_{\boldsymbol{\theta}^1}[R_T^1] = \Omega\left(\frac{\theta_1\theta_{k+1}}{\Delta_{(k)}}\frac{(n-k)}{k}\ln T\right)$ (noting that for instance MNL$(n,\boldsymbol{\theta}^1)$, $\Delta_{(k)}=\epsilon$), which concludes the proof

$\square$

### D.2 Algorithm pseudocode: Top-$k$-regret

---
**Algorithm 3** *Rec-MaxMin-UCB*

---
1: **init:** $\alpha > 0.5$, $\mathbf{W} \leftarrow [0]_{n\times n}$, $\mathcal{B}_0 \leftarrow [\emptyset]_k$
2: **for** $t = 1,2,3,\ldots,T$ **do**
3:     Set $I \leftarrow [n]$, $\mathbf{N} = \mathbf{W} + \mathbf{W}^\top$, and $\hat{\mathbf{P}} = \frac{\mathbf{W}}{\mathbf{N}}$. $N = [n_{ij}]_{n\times n}$ and $\hat{P} = [\hat{p}_{ij}]_{n\times n}$.
4:     Define $u_{ij} = \hat{p}_{ij} + \sqrt{\frac{\alpha\ln t}{n_{ij}}}$, $\forall i,j \in [n], i \neq j$, $u_{ii} = \frac{1}{2}$, $\forall i \in [n]$. $\mathbf{U} = [u_{ij}]_{n\times n}$
5:     **for** $h = 1,2,\ldots,k-1$ **do**
6:         $\mathcal{C}_t^h \leftarrow \{i \in I \mid u_{ij} > \frac{1}{2}, \forall j \in I \setminus \{i\}\}$; $\mathcal{B}_t(h) \leftarrow \mathcal{C}_t^h \cap \mathcal{B}_{t-1}(h)$
7:         **if** $\mathcal{B}_t(h) \neq \emptyset$, **then** set $I \leftarrow I \setminus \mathcal{B}_t(h)$ and $S_t \leftarrow S_t \cup \mathcal{B}_t(h)$
8:         **if** $\mathcal{B}_t(h) = \emptyset$ **then**
9:             **if** $\mathcal{C}_t^h = \emptyset$ **then** $\mathcal{C}_t^h \leftarrow I$; **elseif** $|\mathcal{C}_t^h| > 0$ set $\mathcal{B}_t(h) \leftarrow$ *build_S*$(\mathbf{U}, S_t, I, 1)$
10:           $S_t \leftarrow S_t \cup$ *build_S*$(\mathbf{U}, S_t, I, k - |S_t|)$. Exit and Goto Line 15)
11:         **end if**
12:     **end for**
13:     $\mathcal{C}_t^k \leftarrow \{i \in I \mid u_{ij} > \frac{1}{2}, \forall j \in I \setminus \{i\}\}$; $\mathcal{B}_t(k) \leftarrow \mathcal{C}_t \cap \mathcal{B}_{t-1}(k)$
14:     **if** $|\mathcal{C}_t^k| = 1$, **then** $\mathcal{B}_t(k) \leftarrow \mathcal{C}_t^k$, and set $S_t \leftarrow S_t \cup \mathcal{C}_t^k$; **else** $S_t \leftarrow S_t \cup$ *build_S*$(\mathbf{U}, S_t, I, 1)$
15:     Play $S_t$, and receive: $\boldsymbol{\sigma}_t \in \boldsymbol{\Sigma}_{S_t}^k$
16:     $W(\sigma_t(k'),i) \leftarrow W(\sigma_t(k'),i) + 1$ $\forall i \in S_t \setminus \sigma_t(1:k')$ for all $k' = 1,2,\ldots,k)$
17: **end for**

---

### D.3 Proof of Thm. 8

**Theorem 8** (*Rec-MaxMin-UCB*: High Probability Regret bound)**.** *Given a fixed time horizon $T$ and $\delta \in (0,1)$, with high probability $(1-\delta)$, the regret incurred by Rec-MaxMin-UCB for Top-$k$-regret admits the bound $R_T^k \leq \left(2\left[\frac{2\alpha n^2}{(2\alpha-1)\delta}\right]^{\frac{1}{2\alpha-1}} + 2\bar{D}^{(k)}\ln\left(2\bar{D}^{(k)}\right)\right)\Delta'_{\max} + \frac{4\alpha\ln T}{k}\left(\sum_{b=k+1}^{n}\frac{(\theta_k-\theta_b)}{\hat{D}^2}\right)$, where $D^{(k)}$ is an instance dependent constant (see Lem. 26, Appendix), $\Delta'_{\max} = \frac{\left(\sum_{i=1}^{k}\theta_i - \sum_{i=n-k+1}^{n}\theta_i\right)}{k}$, and $\hat{D} = \min_{g\in[k-1]}(p_{kg} - p_{bg})$.*

*Proof.* For ease of analysis we assume $\theta_1 \geq \theta_2 \geq \ldots \theta_k > \theta_{k+1} \geq \ldots \geq \theta_n$ and hence $S_{(k)} = [k]$.

We use the same notations as introduced in the proof of Thm. 5. Note that Lem. 15 holds in this case as well. So that concludes the **Random-Exploration** phase.

**Analysis of the Progress phase:** We next proceed to analyse the **Progress** phase from round $f(\delta)+1$ to $T_0(\delta)$, where $T_0(\delta)$ is defined to be such that

$$T_0(\delta) = \arg\min_{t>f(\delta)} \mathcal{B}_t = [k]$$

The goal of this phase is to show that the length of interval $[f(\delta) + 1, T_0(\delta)]$ is *'small'*, precisely $T_0(\delta) \le 2f(\delta) + 2\bar{D}^{(k)} \ln(2\bar{D}^{(k)})$, where $\bar{D}^{(k)} := \sum_{r=1}^{k} D^{(r)}$, and $D^{(r)} := \sum_{i \in [n]} \sum_{j \in W(r)} D_{ij}^r$ (see Lem. 26). Note here $\bar{D}^{(k)}$ is a problem dependent constant, independent of $T$.

**Notations:** We first define few notations for ease of analysis: Let us first define the set of items $W(i) = \{ j \in [k] \mid \theta_j > \theta_i \}$ be the set of items in the *Top-k Best-Items* strictly better than $i$, and $Z(i) = \{j \in [k] \mid \theta_j < \theta_i\}$ be the set of items in $[k]$ worse than item $i$, for some $i \in [k]$.

For any $g \in [k]$,

$$D_{ij}^g = \begin{cases} \frac{4\alpha}{\min((p_{gi}-\frac{1}{2})^2,(p_{gj}-\frac{1}{2})^2)} \text{ if } i,j \in Z(g) \\ \frac{4\alpha}{(p_{gi}-p_{ji})^2} \text{ if } i \in W(g) \text{ and } j \in Z(g) \\ D_{gj}^g \text{ if } \theta_i = \theta_g \end{cases}, D_{ij}^g = D_{ji}^g$$

for any pair $(i, j) \in [n] \times [n]$, $i \ne g$, $j \ne g$. and $D_{gi}^g = \frac{4\alpha}{(p_{gi}-\frac{1}{2})^2}$, where $p_{ij} = \frac{1}{2} + \frac{\theta_i - \theta_j}{2(\theta_i + \theta_j)}$ for all $i, j \in [n]$.

Towards analysing the **Progress** phase we first make the following key observations:

- **Observation 1:** At any round $t \in [T]$, $\mathcal{B}_t(i)$ is either singleton or an empty set, for all $i \in [k]$, which follows by the construction of $\mathcal{B}_t(i)$.

- **Observation 2:** For any item $i \in [k]$ in the *Top-k Best-Items*, at any round $t > f(\delta)$ if $j \in \mathcal{B}_t(|W(j)| + 1 : k - |Z(j)|)$ for all $j \in W(i)$, and $i \in \mathcal{B}_t$ such that $\mathcal{B}_t(x) = i$ for some $x \in \{|W(i)| + 1 \ldots k - |Z(i)|\}$ and for any $x' \in \{|W(i)| + 1, \ldots, (x - 1)\}$, $\theta_{\mathcal{B}_t(x')} = \theta_i$, then $\mathcal{B}_{t'}(x) = \{i\} \forall t' > t$— in other words $i$ will continue to reside in slot $\mathcal{B}_{t'}(x)$ for any $t' > t$.

  We next define another notation $T_0^i(\delta)$ for any $i \in [k]$ such that $T_0^i(\delta) = \arg\min_{t>f(\delta)}\{\forall j \in W(i), j \in \mathcal{B}_t(|W(j)| + 1 : k - |Z(j)|), i = \mathcal{B}_t(x), \text{and} \forall x' \in \{|W(i)| + 1, \ldots, (x - 1)\}, \theta_{\mathcal{B}_t(x')} = \theta_i\}$. Clearly $\max_{i \in [k]} T_0^i(\delta) = T_0(\delta)$ as defined above.

- **Observation 3:** $T_0(\delta)$ is not far from $f(\delta)$—we prove this in a stepwise manner, to explain it in an intuitive level assume $\theta_1 > \ldots > \theta_k$, Then we first show that $T_0^1(\delta)$ is bounded. Once item 1 is secured in its slot $\mathcal{B}_{T_0^1(\delta)}(1)$, we proceed to bound $T_0^2(\delta)$, and so on till $T_0^k(\delta) = T_0(\delta)$ (see Lem. 26 for the formal details which holds due to Lem. 24 and 25).

We find it convenient to define one more definition before proving Lem. 26:

**Definition 23** ($g$-Unsaturated Pairs). At any time $t \in [T]$, for any item $g \in [k]$ and any pair of two distinct pair of items $i, j \in [n]$, we call the pair $(i, j)$ to be *$g$-unsaturated* at time $t$ if $n_{ij}(t) \le D_{ij}^g \ln t$. Otherwise, we call the pair *$g$-saturated* at $t$.

**Lemma 24.** *Assuming $\forall t > f(\delta)$, and $\forall i, j \in [n]$ $p_{ij} \in [l_{ij}(t), u_{ij}(t)]$, for some $\delta \in (0, 1)$: At any iteration $t > f(\delta)$, for any $g \in [k]$, if $\exists$ an item $i \in Z(g)$, i.e. $\theta_g > \theta_i$ and both $i, j \in \mathcal{C}_t^h$ for some $h \in [k]$, then the pair $(g, i)$ is $g$-unsaturated at $t$.*

*Proof.* By assumption $\forall i, j \in [n]$ $p_{ij} \le u_{ij}(t)$. Now if $\exists h \in [k]$ such that a pair $(g, i)$ such that both $i, g \in \mathcal{C}_t^h$, then it has to be the case that $u_{gi} > \frac{1}{2}$ and $u_{ig} > \frac{1}{2}$.

But then suppose $(g, i)$ was indeed $g$-saturated at time $t$, i.e. $n_{1i}(t) > D_{gi}^g \ln t$, this implies:

$$u_{ig}(t) = \hat{p}_{ig}(t) + c_{ig}(t) \le p_{ig}(t) + 2c_{ig}(t) = p_{ig}(t) + \Delta_i^g = \frac{1}{2},$$

which implies there cannot exist $i \notin \mathcal{C}_t^h$ if $g \in \mathcal{C}_t^h$ for any $h$, which leads to a contradiction. Hence the pair $(g, i)$ must be $g$-unsaturated at $t$. $\square$

**Lemma 25.** *Assuming $\forall t > f(\delta)$, and $\forall i, j \in [n]$ $p_{ij} \in [l_{ij}(t), u_{ij}(t)]$, for some $\delta \in (0,1)$. Consider any $g \in [k]$. At any iteration $t > f(\delta)$, for any set $S_t \not\ni g$, $0 \leq |S_t| < k$ if $\exists$ an item $a \in Z(g)$, i.e. $\theta_g > \theta_a$, such that $a = \arg\max\limits_{c \in I \setminus S_t} \left[\min_{i \in S_t} u_{ci}(t)\right]$, then $\exists$ atleast one item $b \in S$ such that the pair $(a,b)$ is unsaturated at $t$.*

*Proof.* Firstly the important observation to make is at any round $t$, and in any of its sub-phase $h \in [k]$, our set building rule ensures that $u_{ji} > \frac{1}{2}$ for $j \in S_t$ and $i \notin S_t$.

Moreover since $a = \arg\max\limits_{c \in I \setminus S_t} \left[\min_{i \in S_t} u_{ci}(t)\right]$ and $a \notin S_t$, there must exist an item $b$ in $S$ such that $u_{ab}(t) > u_{gb}(t)$ as otherwise $g$ would have been picked instead of $a$. But following the argument above we also know that $u_{ba}(t) > \frac{1}{2}$. Now $b$ can fall into the following three categories:

**Case-1** $\left[b \in Z(g)\right]$: We first note that:

$$u_{ba}(t) - l_{ba}(t) = u_{ba}(t) + u_{ab}(t) - 1 > \frac{1}{2} + u_{gb}(t) - 1 > \frac{1}{2} + p_{gb}(t) - 1 = p_{gb} - \frac{1}{2},$$

but on the other hand if the pair $(a,b)$ is indeed $g$-saturated at $t$, i.e. $n_{ba}(t) > D_{ba}^g \ln t$. Then we have that

$$u_{ba}(t) - l_{ba}(t) = 2c_{ba}(t) \leq \sqrt{\min\left(\left(p_{gb} - \frac{1}{2}\right)^2, \left(p_{ga} - \frac{1}{2}\right)^2\right)} \leq \left(p_{gb} - \frac{1}{2}\right).$$

**Case-2** $\left[b \in W(g)\right]$:

In this case suppose if the pair $(a,b)$ is indeed $g$-saturated at $t$, i.e. $n_{ba}(t) > D_{ba}^g \ln t$ we have

$$2c_{ab}(t) \leq \sqrt{\left(p_{gb} - p_{ab}\right)^2} \leq \left(p_{gb} - p_{ab}\right).$$

It is important to note that the right hand side of the above inequality is positive since for this case $\theta_b > \theta_g > \theta_a$. But this implies

$$u_{ab}(t) \leq p_{ab}(t) + 2c_{ab}(t) = p_{gb} + 2c_{ab}(t) - (p_{gb} - p_{ab}) \leq p_{gb} < u_{gb}(t)$$

which leads to a contradiction again.

**Case-3** $\left[b : \theta_b = \theta_g\right]$: The analysis in this case goes similar to **Case-2** above which finally leads to the contradiction that $u_{ab} < p_{gb} = \frac{1}{2} < u_{gb}(t)$.

Hence combining the above three cases, it follows that the unless the pair $(b,a)$ is $g$-unsaturated at round $t$, $a \in Z(g)$ can not show up prior to $g$. $\qquad\square$

**Assumption:** Recall we assumed $\theta_1 \geq \theta_2 \geq \ldots \geq \theta_k$. For ease of explanation (without loss of generality by relabelling the items) we also assume that at any time $t$, for any pair of items $(i,j)$ such that $i, j \in [k]$, $\theta_i = \theta_j$, $i < j$, and if it happens to be the case that both $i, j \in \mathcal{B}_t$, with $\mathcal{B}_t(x) = i$, $\mathcal{B}_t(y) = j$ then $x < y$.

**Lemma 26.** *Assume $\forall t > f(\delta)$, and $\forall i, j \in [n]$ $p_{ij} \in [l_{ij}(t), u_{ij}(t)]$, for some $\delta \in (0,1)$. Then if we define $T_0(\delta)$ such that: $T_0(\delta) = \min\{t > f(\delta) \mid \mathcal{B}_t = [k]\}$, it can be upper bounded as $T_0(\delta) \leq 2f(\delta) + 2\bar{D}^{(k)} \ln\left(2\bar{D}^{(k)}\right)$, where $\bar{D}^{(k)} := \sum_{r=1}^k D^{(r)}$, and $D^{(r)} := \sum_{i \in [n]} \sum_{j \in W(r)} D_{ij}^r$ (recall the rest of the notations as defined above).*

*Proof.* Combining Lem. 24 and 25 we first aim to bound the term $T_0^1(\delta)$ such that the first time after $T_0(\delta)$, when $\mathcal{B}_t(1) = 1$ and post which it follows that $\mathcal{B}_t = \{1\}$ for all $t > T_0(\delta)$.

**Bounding $T_0^1(\delta)$** : Note that for any $t > f(\delta)$, $Z(1) = n - 1$ always. So the only way 1 can miss the $\mathcal{B}_t(1)$ slot is if $\exists i \neq 1$ and $\theta_1 > \theta_i$ (due to the relabelling **Assumption** above) which occupies $\mathcal{B}_t(1)$. All we need to figure out is the worst possible number of rounds *Rec-MaxMin-UCB* would take to 1-saturate all the pairs, precisely $\sum_{i<j} D_{ij}^1 \ln t$ many pairwise updates should be done within $t$ rounds. Now using a similar chain of argument given in Lem. 20 (along with Lem. 24 and 25), since any round $t > f(\delta)$ updates atleast one 1-unsaturated pair, we find that

$$T_0^1(\delta) = \min\{t > f(\delta) \mid t > f(\delta) + D^{(1)} \ln t\},$$

where $D^{(1)} := \sum_{i \in [n]} \sum_{j \in W(1)} D_{ij}^1$

**Bounding $T_0^2(\delta)$**: Note that once $1 \in \mathcal{B}_t(1)$, for any $t > T_0^1(\delta)$, $1 \in S_t$. And also either $\theta_2 = \theta_1$ or $\theta_2 < \theta_1$. But in either case $2 \in \mathcal{C}_t^2$, as $u_{2j} > \frac{1}{2}$ for all $j \in [n] \setminus \{1\}$. Then the only way the one can stop 2 occupying the slot $\mathcal{B}_t(2)$ is if there exists some other item $i \neq 2$ and $\theta_2 > \theta_i$ (due to the relabelling **Assumption** above) which occupies $\mathcal{B}_t(2)$. But then the algorithm picks $\mathcal{B}_t(2)$ in $S_t$, i.e. $i \in S_t$ alongside 1 and it get compared with 1 at each round it is picked. Moreover the last element of $S_t$ is always picked by the *build_S* subroutine, so following the three case analyses of Lem. 25, the maximum number of rounds till which 2 can miss the slot $\mathcal{B}_t(2)$ is

$$T_0^2(\delta) = \min\{t \mid t > T_0^1(\delta) + D^{(2)} \ln t\},$$

where $D^{(2)} := \sum_{i \in [n]} \sum_{j \in W(2)} D_{ij}^2$.

Following the same argument we can state a general result that for any $r \in [k] \setminus \{1\}$:

**Bounding $T_0^r(\delta)$** where $D^{(r)} := \sum_{i \in [n]} \sum_{j \in W(r)} D_{ij}^r$:

$$T_0^r(\delta) = \min\{t \mid t > T_0^{r-1}(\delta) + D^{(r)} \ln t\}$$

Then combining above for $r = 1, \ldots k$ we get $T_0(\delta) = T_0^k(\delta)$ should be such that

$$T_0^k(\delta) = \min\{t \mid t > f(\delta) + \sum_{r=1}^{k} D^{(r)} \ln t\}$$

And now following the exact same analysis of Lem. 20, it is easy to see that the above inequality satisfies for $t = 2f(\delta) + 2\sum_{r=1}^{k} D^{(r)} \ln\left(\sum_{r=1}^{k} D^{(r)}\right)$. So that bounds $T_0(\delta) \leq 2f(\delta) + 2\sum_{r=1}^{k} D^{(r)} \ln\left(\sum_{r=1}^{k} D^{(r)}\right)$.

$\square$

*Analysis for* **Saturation** *phase* $(t > T_0(\delta))$:

- **Observation** 4**:** After $T_0(\delta)$, now $\mathcal{B}_t = S_{(k)} = [k]$, with $\mathcal{B}_t(k) = \theta_k$, and thereafter i.e. $\mathcal{B}_t = [k]$ for all $t > T_0(\delta)$. Note that in this phase the algorithm always plays either $S_t = [k]$ or it plays $S_t = [k-1] \cup \{b\}$, $b \notin [k]$. So for any $t > T_0(\delta)$, $S_t \cap \mathcal{B}_t = [k-1]$. Then at any time $t$ is a suboptimal item $b \in [n] \setminus [k]$ comes in $S_t$ then it gets compared to all items in $[k-1]$ (owing to *Rank-Breaking*). But can not happen for too long and after a time $\mathcal{C}_t \cap \{b\} = \emptyset$, when the algorithm *Rec-MaxMin-UCB* will not play $b$ any more. This holds true for any $b \in [n] \setminus [k]$, for which the algorithm will left with no other choice for $S_t$ other than $S_t = [k]$ when it incurs no regret. See Lem. 27 and 28 for the formal claims.

**Lemma 27.** *Assume $\mathcal{B}_t = S_{(k)} = [k]$ for all $t > T_0(\delta)$. Then the total cumulative regret of Rec-MaxMin-UCB post $T_0(\delta)$ is upper bounded by:*

$$R_T^k(T_0(\delta) : T) := \sum_{t=T_0(\delta)}^{T} r_t^k \leq \frac{4\alpha \ln T}{k}\left(\sum_{b=k+1}^{n} \frac{(\theta_k - \theta_b)}{\hat{D}^2}\right),$$

*where $\hat{D} = \min_{g \in [k-1]}(p_{kg} - p_{bg})$.*

*Proof.* Note that when $\mathcal{B}_t = S_{(k)} = [k]$, at any such round *Rec-MaxMin-UCB* plays the set $S_t$ such that the first $(k-1)$ items of $\mathcal{B}_t$ are always included in $S_t$, i.e. $|S_t \cap \mathcal{B}_t(1:k-1)| = k-1$. The $k^{th}$ element of $\mathcal{B}_t$ only gets replaced by a suboptimal element $b \in [n] \setminus [k]$ only if $\exists g \in \mathcal{B}_t(1:k-1)$ such that the pair $(g, b)$ is *unsaturated*, in a sense that $u_{bg} > u_{kg}$, and hence $b$ got picked by the algorithm instead of $k$ (in Line 14).

But is that possible for long? Precisely, we now show that any such suboptimal item $b \in [n] \setminus [k]$ can not get selected by the algorithm for more than $\frac{4\alpha \ln T}{\hat{D}^2}$ times. This is since for any $S_t = [k-1] \cup \{b\}$ (recall that $\mathcal{B}_t(1:k-1) = [k-1]$), once played for $\frac{4\alpha \ln T}{\hat{D}^2}$ times (say this happens at time $t = \tau$), we know that $\forall g \in [k-1]$ such that number of times the pair $(g, b)$ gets updated is exactly $\frac{4\alpha \ln T}{\hat{D}^2}$ too due to *Rank-Breaking* on Top-$k$-ranking Feedback. But this implies for any $g \in [k-1]$,

$$u_{bg}(\tau) \le p_{bg} + 2c_{bg}(\tau) = p_{kg} + 2c_{bg}(\tau) - (p_{kg} - p_{bg}) \le p_{kg} \le u_{kg}(\tau) \qquad (18)$$

where the first and last inequality follows by definition of $u_{bg}$ and Lem. 15, the second last inequality follows due to the fact that since $n_{gb}(\tau) \ge \frac{4\alpha \ln T}{\hat{D}^2}$

$$2c_{gb}(\tau) \le 2\sqrt{\frac{\alpha \ln T}{\frac{4\alpha \ln T}{\hat{D}^2}}} = \hat{D} \le (p_{kg} - p_{bg}),$$

since by definition $\hat{D} = \min_{g \in [k-1]}(p_{kg} - p_{bg})$. Then Eqn. 18 leads to a contradiction showing $u_{bg}(\tau) < u_{kg}(\tau) \implies b$ can not replace $k$ at any round $t > \tau$.

The rest of the analysis simply follows from the fact that since any $b \in [n] \setminus [k]$ can appear for only $\left(\frac{4\alpha \ln T}{\hat{D}^2}\right)$ times and it replaces the item $\mathcal{B}_t(k) = k$, hence the cost incurred for $b$ is $\frac{(\theta_k - \theta_b)}{k}$ (by Eqn. 2.2). Thus the total regret incurred in saturation phase is $\frac{4\alpha \ln T}{k}\left(\sum_{b=k+1}^{n} \frac{(\theta_k - \theta_b)}{\hat{D}^2}\right)$. $\qquad \square$

**Lemma 28.** *For any $\delta \in (0,1)$, with probability at least $(1-\delta)$, the total cumulative regret of Rec-MaxMin-UCB can be upper bounded by:*

$$R_T^k \le \left(2f(\delta) + 2\bar{D}^{(k)} \ln\left(2\bar{D}^{(k)}\right)\right)\Delta'_{\max} + \frac{4\alpha \ln T}{k}\left(\sum_{b=k+1}^{n} \frac{(\theta_k - \theta_b)}{\hat{D}^2}\right)$$

*where $\Delta'_{\max} = \frac{(\sum_{i=1}^{k} \theta_i - \sum_{i=n-k+1}^{n} \theta_i)}{k}$, and $f(\delta)$, $\bar{D}^{(k)}$ and $\hat{D}$ is as defined in Lem. 15, 26, 27.*

*Proof.* This can be proved just by combining the claims of Lem. 26 and 27. Note from Lem. 26 that till the **Progress** phase $T_0(\delta)$, the algorithm can play any arbitrary sets $S_t$ for which the maximum regret incurred can be $\Delta'_{\max} = \frac{(\sum_{i=1}^{k} \theta_i - \sum_{i=n-k+1}^{n} \theta_i)}{k}$. Thereafter the algorithm enters into **Saturation** phase at which the maximum regret in can incur is $\frac{4\alpha \ln T}{k}\left(\sum_{b=k+1}^{n} \frac{(\theta_k - \theta_b)}{\hat{D}^2}\right)$ as follows from Lem. 28, which concludes the proof. $\qquad \square$

The entire analysis above thus concludes the proof of Thm. 8. $\qquad \square$

## Proof of Thm. 9

**Theorem 9.** *The expected regret incurred by MaxMin-UCB for Top-k-regret is:*

$$\mathbf{E}[R_T^1] \le \left(2\left[\frac{2\alpha n^2}{(2\alpha-1)}\right]^{\frac{1}{2\alpha-1}} \frac{2\alpha-1}{\alpha-1} + 2\bar{D}^{(k)} \ln\left(2\bar{D}^{(k)}\right)\right)\Delta'_{\max} + \frac{4\alpha \ln T}{k}\left(\sum_{b=k+1}^{n} \frac{(\theta_k - \theta_b)}{\hat{D}^2}\right).$$

*Proof.* The proof essentially follows same as the proof of Thm. 6 by integrating the $\delta$ dependent term $\left[\frac{2\alpha n^2}{(2\alpha-1)\delta}\right]^{\frac{1}{2\alpha-1}}$ in $R_T^k$ (see Thm. 8) from $\delta = 0$ to $\infty$. $\qquad \square$

# E Experiment Details

We report numerical results of the proposed algorithms run on the following $\mathrm{MNL}(n, \boldsymbol{\theta})$ models:

**$\mathrm{MNL}(n, \boldsymbol{\theta})$ Environments.** 1. *g1*, 2. *g4*, 3. *arith*, 4. *geo* all with $n = 16$ and two larger models 5. *arith-big*, and 6. *geo-big* each with $n = 50$ items. Their individual score parameters are as follows: **1. g1:** $\theta_1 = 0.8$, $\theta_i = 0.2$, $\forall i \in [16] \setminus \{1\}$ **2. g4:** $\theta_1 = 1$, $\theta_i = 0.7$, $\forall i \in \{2, \ldots 6\}$, $\theta_i = 0.5$, $\forall i \in \{7, \ldots 11\}$, and $\theta_i = 0.01$ otherwise. **3. arith:** $\theta_1 = 1$ and $\theta_i - \theta_{i+1} = 0.06$, $\forall i \in [15]$. **4. geo:** $\theta_1 = 1$, and $\frac{\theta_{i+1}}{\theta_i} = 0.8$, $\forall i \in [15]$. **5. har:** $\theta_1 = 1$ and $\theta_i = 1 - \frac{1}{i}$, $\forall i \in \{2, 3, \ldots, 16\}$. **6. arithb:** $\theta_1 = 1$ and $\theta_i - \theta_{i+1} = 0.02$, $\forall i \in [49]$. **7. geob:** $\theta_1 = 1$, and $\frac{\theta_{i+1}}{\theta_i} = 0.9$, $\forall i \in [49]$.