[Reviews · NeurIPS 2019]

Reviewer 1



The authors study the problem of online learning with subset-wise preferences with relative feedback information under the Multinomial logit choice model. More speci_x000C_cally, they study two regret minimisation problems over subsets of a fi_x000C_nite ground set [n]. In the fi_x000C_rst setting, the learner can play subsets of size bounded by a maximum size and receives top-m rank-ordered feedback. For this setting, they give a lower bound and an instance-dependent and order-optimal winner-regret algorithm with regret O( n/m \ln T) based on the well-known upper-con_x000C_dence bound (UCB) strategy. In the second setting the learner can play subsets of a _x000C_xed size k with a full subset ranking observed as feedback, they also derive a regret lower bound, and then give a UCB-based algorithm, along with a matching upper bound regret of O(n/k \ln T). The authors also provide an experimental study with synthetically generated data to validate their _x000C_findings. The authors give a thorough study of interesting extensions of the dueling bandit problem, including deriving lower bounds, and giving algorithms with theoretical guarantees. The paper has a good quality and is written in a good way, and should be signifi_x000C_cant for the preference- based learning community.

Reviewer 2



For me, the regrets they introduced seem unnatural. In their feedback settings, the best arm identification framework is more appropriate. Show some cases in which the introduced regrets are more appropriate. [comments after authors' feedback] I understood that their regret definition is a natural extension of the popular dueling bandit regret. However, I still think that regret minimization setting is unnatural considering real applications.

Reviewer 3



The paper considers a bandit problem where the learner selects a subset of (up to) k>= 2 arms in each round. Subsequently, the learner observes as feedback a rank-ordered list of m >=1 items from the subset, generated probabilistically according to the Pluckett-Luce distribution on rankings based on the multinomial logit choice. That is, each arm is associated with a weight and, at each round, the online algorithm plays a subset S of at most k arms. It then receives feedback according to one of the following models: 1. Winner Feedback: The environment returns one item from S. The probability of each subset is proportional to its weight. This scheme is known as the Placket-Luce probability model" on S. 2. Top-m Ranking Feedback: The environment returns an ordered list of m items sampled without replacement from the Plackett-Luce probability model on S. The main contribution of the author(s) is to give algorithms for regret minimization that perform optimally (up to constants) under the assumptions of the feedback models (1) and (2). The exact bounds on the regret are somewhat involved to state here, but they depend logarithmically on the number of rounds and, interestingly (especially in terms of upper bound), on the weight distribution over arms. In terms of techniques, while I am not an expert in the area so I cannot judge their novelty, I did find the proposed algorithms and their analysis interesting and intuitive. A possible criticism is that the setting considered by the author(s) is similar to submitted Paper 396 (indeed, it was flagged as a possible dual submission). That said, the two papers seem to use sufficiently different techniques. Overall, I found the paper interesting. My main concern has to do with whether the assumptions of the proposed models are natural enough, and their relevance to practice.

[Author Response · NeurIPS 2019]

1. We thank all the reviewers for their careful readings and constructive comments.

2. **Reviewer #1**: Many thanks for appreciating our work.

3. **Reviewer #2 (Re. Motivation behind the Regret definitions):**

4. Note that following the RUM interpretation of MNL model (please see response to Rev. #3 for details), the score
5. parameter $\theta_i$ of each item $i \in [n]$ essentially represents the mean utility/reward of item $i$, which in turn governs its
6. preference relation w.r.t. the other items (based on the "Feedback type", Sec. 2.1). Thus our regret definitions (i.e.
7. both *Winner* and *Top-k regret*, Sec 2.2), simply penalize the learner for pulling any suboptimal set in terms of the
8. sub-optimality in its average utility-score w.r.t. that of the optimal set (which is $a^*$ for *Winner regret*, and $S_{(k)}$ for *Top-k*
9. *regret*) — an intuitive quantification of loss/value of a subset in terms of the underlying utility scores of its items.

10. Re. Applications: As discussed in the Introduction, some motivating applications of our problem lies in various kind of
11. *partial monitoring frameworks*, e.g. launching new products, recommender systems, crowdsourcing etc., where value
12. of a subset is measured in terms of the average utility-scores ($\theta_i$s) of its items, but the learner only gets to observe a
13. preference feedback of the selected items drawn according to the MNL($\boldsymbol{\theta}$) model, $\boldsymbol{\theta} = (\theta_1, \ldots, \theta_n)$.

14. Moreover, as we clarified in Rem. 1 and 2, for the special case of only two-sized subsets (i.e. when $k = 2$), our regret
15. definition simply boils down to that of '*Dueling Bandit*' problem – an extensively studied and well accepted notion of
16. regret in bandit-literature (Ref. [5,12,40-47]), which too is based on the concept of penalizing every subset (i.e. pair of
17. items as $k = 2$) in terms sub-optimality of average item scores. In fact, the very few recent works that extends Dueling
18. Bandits to subsetwise feedback (*Multi-Dueling bandits*), also use the same notion of regret as ours (see Ref. [11,39]).

19. **Reviewer #3 (Re. Assumptions of the proposed models and practical relevance):**

20. We have assumed Multinomial Logit (MNL) (alternatively known as Plackett Luce) McFadden and Train [2000], Luce
21. [1959] as our subsetwise feedback model which is a widely used preference model in econometrics and social choice
22. theory literature (Ref. [7],Soufiani et al. [2013]), specially for assortment selection problems (Refs. [2,3,4]), as well as
23. in machine learning community, be that offline batch optimization (Ref. [23,29,39]), or online learning setting (Ref.
24. [17,35, 40]) etc. In fact, even for the special case when subsetsize $k = 2$, the model is extensively studied as *Bradley*
25. *Terry Luce* (BTL) model Negahban et al. [2012], Rajkumar and Agarwal [2014], Shah and Wainwright [2015], and its
26. various extensions have also been considered Wen and Koppelman [2001], Yan et al. [2019] — thus MNL model is
27. indeed one of the most well studied preference model, which has natural applications to various real world scenarios,
28. e.g. customer preferences, recommender systems, voting methods, or more generally any application which aims to
29. aggregate information from preferences over discrete choices. (see response to Rev. #2 for more applications).

30. For a more theoretical interpretation of MNL feedback model (Def. 1): MNL model belongs to the class of Random
31. Utility Models(RUM), which assumes an underlying utility scores of the items $\theta_i' \in \mathbf{R}$ for each item $i \in [n]$, and assigns
32. a conditional distribution $\mathcal{D}_i(\cdot|\theta_i')$ for scoring item $i$. Upon receiving any subset $S \subseteq [n]$, the environment first draws a
33. random utility score $X_i \sim \mathcal{D}_i(x_i|\theta_i')$ for each item $i \in S_t$, and selects the winner item $J = j$ with probability of $X_j$
34. being the maximum among all the scores of items in $S$, i.e. Winner Feedback: $Pr(J = j) \sim Pr(X_j > X_{j'} \ \forall j' \in$
35. $S \setminus \{j\}) \ \forall j \in S$. Now it can be shown that when $\mathcal{D}_i'$s are Gumbel($\theta_i, 1$) distributions (Ref. [7],Soufiani et al. [2013]),
36. i.e. $\mathcal{D}_i(x_i|\theta_i') = e^{(x_j - \theta_j')} e^{-e^{(x_j - \theta_j')}}$, then $Pr(i|S_t) := Pr(X_i > X_j \ \forall j \in S_t \setminus \{i\}) = \frac{e^{\theta_i'}}{\sum_{j \in S_t} e^{\theta_j'}}$ — which precisely
37. gives rise to the MNL choice model. (We used $\theta_i = e^{\theta_i'}, \ \forall i \in [n]$. Unfortunately due to space constraints we could not
38. include this RUM interpretation of MNL model, which really sheds light into its specific mathematical form.)

39. We sincerely request the reviewers to kindly reconsider their scores based on the above clarifications.

# References

41. R Duncan Luce. *Individual Choice Behavior: A Theoretical Analysis*. Wiley, 1959.

42. Daniel McFadden and Kenneth Train. Mixed mnl models for discrete response. *Journal of applied Econometrics*, 2000.

43. Sahand Negahban, Sewoong Oh, and Devavrat Shah. Iterative ranking from pair-wise comparisons. In *Advances in Neural*
44. *Information Processing Systems*, 2012.

45. Arun Rajkumar and Shivani Agarwal. A statistical convergence perspective of algorithms for rank aggregation from pairwise data.
46. In *Proceedings of 31st International Conference on Machine Learning*, 2014.

47. Nihar B Shah and Martin J Wainwright. Simple, robust and optimal ranking from pairwise comparisons. *arXiv preprint*
48. *arXiv:1512.08949*, 2015.

49. Hossein Azari Soufiani, David C Parkes, and Lirong Xia. Preference elicitation for general random utility models. In *Uncertainty in*
50. *Artificial Intelligence*, page 596. Citeseer, 2013.

51. Chieh-Hua Wen and Frank S Koppelman. The generalized nested logit model. *Transportation Research Part B: Methodological*, 35
52. (7):627–641, 2001.

53. Yongnan Yan, Xiangdong Xu, and Anthony Chen. Is it necessary to relax the iid assumptions in the logsum-based accessibility
54. analysis? *Transportation Research Record*, page 0361198119839972, 2019.


[Meta-Review · NeurIPS 2019]

Although the reviewers have concerns about practical relevance of the setting, they agree the paper is theoretically interesting. In the introduction and in their response, the authors also motivate the setting by explaining how it generalizes the popular dueling bandit setting and how it has previously been studied by [11,39].